# Learning Harmonized Representations for Speculative Sampling

**Lefan Zhang, Xiaodan Wang, Yanhua Huang,**[*] **Ruiwen Xu**
Xiaohongshu Inc.
Shanghai, China
{lefan,xiaodan2,yanhuahuang,ruiwenxu}@xiaohongshu.com

## Abstract

Speculative sampling is a promising approach to accelerate the decoding stage for Large Language Models (LLMs). Recent advancements that leverage target LLM's contextual information, such as hidden states and KV cache, have shown significant practical improvements. However, these approaches suffer from inconsistent context between training and decoding. We also observe another discrepancy between the training and decoding objectives in existing speculative sampling methods. In this work, we propose a solution named HArmonized Speculative Sampling (HASS) that learns harmonized representations to address these issues. HASS accelerates the decoding stage without adding inference overhead through harmonized objective distillation and harmonized context alignment. Experiments on four LLaMA models demonstrate that HASS achieves 2.81x-4.05x wall-clock time speedup ratio averaging across three datasets, surpassing EAGLE-2 by 8%-20%. The code is available at https://github.com/HArmonizedSS/HASS.

## 1 Introduction

Generative Large Language Models (LLMs), such as GPT-4 (Achiam et al., 2023) and LLaMA (Touvron et al., 2023), have demonstrated remarkable capabilities across a wide range of tasks. Nevertheless, efficiently decoding from these models poses a significant challenge due to the inherent auto-regressive decoding mechanism, which restricts their applicability in time-sensitive scenarios. Speculative sampling (Chen et al., 2023; Leviathan et al., 2023) offers a solution by leveraging additional resources to increase concurrency. Specifically, it employs an efficient draft model to generate draft tokens auto-regressively, which are then concurrently verified by the target LLM. Based on the verification results, a subset of draft tokens that preserves the same distribution as the target LLM is accepted as the final output.

Leviathan et al. (2023) show that the practical performance of speculative sampling is highly related to two factors: the decoding cost of the draft model and its alignment with the target LLM. To develop efficient draft models that are well-aligned with the target LLM, previous works propose to leverage the target LLM's contextual information (Xiao et al., 2024; Li et al., 2024b;c; Du et al., 2024). For instance, EAGLE (Li et al., 2024b;c) employs previous hidden states of the target LLM as the draft model's input features. However, these approaches introduce inconsistent context between training and decoding, as illustrated in Figure 2. During training, the draft model always has access to the target LLM's hidden states in previous timesteps. However, during decoding, the draft model cannot access the target LLM's hidden states for unverified timesteps, resulting in a context misalignment between training and decoding. This issue can be viewed as a form of exposure bias (Bengio et al., 2015; Wang & Sennrich, 2020) at the feature level in speculative sampling.

Another discrepancy is also observed between the objectives of the training and decoding stages. During the decoding stage, the objective of the draft model is to propose tokens that the target LLM is likely to assign high probabilities to (Li et al., 2024c; Miao et al., 2024; Sun et al., 2024). In this scenario, the draft model should focus more on recalling the desired tokens, while the specific

---

[*]Corresponding author.

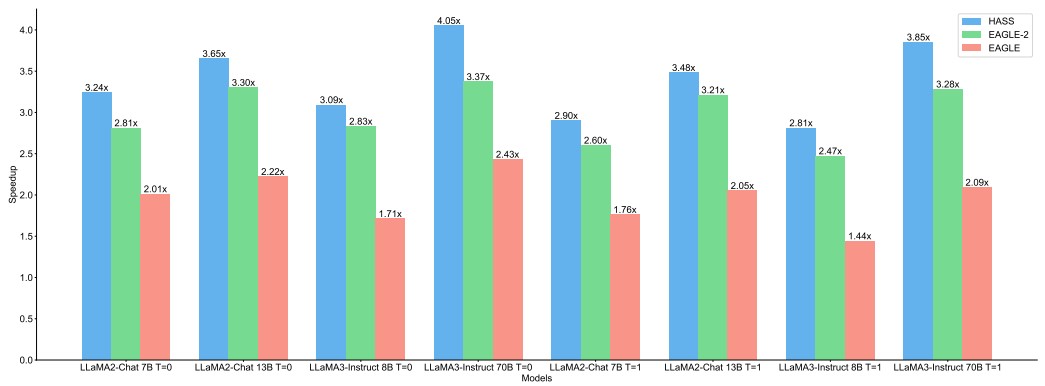

Figure 1: Speedup ratios of different methods on LLaMA2-Chat 7/13B and LLaMA3-Instruct 8/70B with temperature T ∈ {0, 1}, averaging over MT-bench, HumanEval, and GSM8K datasets.

order of these tokens can be somewhat de-emphasized. Moreover, most LLM applications perform nucleus sampling (Holtzman et al., 2020) or top-k sampling (Fan et al., 2018). For these decoding objectives, tokens with high probabilities play a more significant role in determining the output. Therefore, to develop efficient draft models, their training objectives should consider these properties encountered in the decoding stage. To the best of our knowledge, previous works on training draft models for speculative sampling have largely overlooked these decoding considerations.

In this paper, we introduce HArmonized Speculative Sampling (HASS), a novel approach designed to address the aforementioned issues by learning harmonized representations. Specifically, to make draft models aware of the decoding strategy, HASS extends the idea of ranking distillation (Tang & Wang, 2018) from the recommender system to speculative sampling, resulting in a distillation loss focused on the most probable tokens within the target distribution. To mitigate the previously discussed context misalignment between training and decoding, HASS employs a context-aligned training strategy. Together, these two strategies of HASS improve the acceleration performance without any inference overhead and maintain training efficiency.

We conduct experiments across dialogue, code generation, and mathematical reasoning tasks using the MT-bench, HumanEval, and GSM8K datasets, respectively. Building with EAGLE-2 (Li et al., 2024c), HASS achieves 8%-16% acceptance length improvement over it on LLaMA2-Chat 7/13B and LLaMA3-Instruct 8/70B, resulting in 2.81x-4.05x wall-clock time acceleration compared with the vanilla inference on NVIDIA H800 GPU.

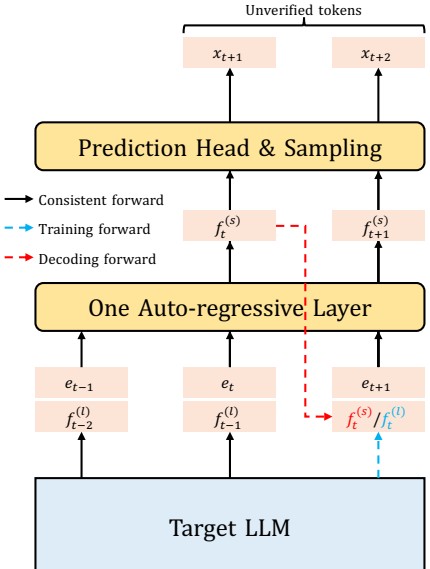

Figure 2: We use EAGLE (Li et al., 2024b) as an example to illustrate the context misalignment, where the speculation starts from timestep $t$. $f^{(l)}$ and $f^{(s)}$ represent hidden states from the target LLM and the draft model. When decoding draft token $x_{t+2}$, the input context is inconsistent between training and decoding.

## 2  PRELIMINARY

**Speculative sampling** leverages the concept of speculative execution (Kung & Robinson, 1981; Hennessy & Patterson, 2011) to reduce wall-clock time from more concurrency. Specifically, given the target LLM $\mathcal{M}^{(l)}$ that is the focus of acceleration, speculative sampling employs a draft model

$\mathcal{M}^{(s)}$ to speculatively and efficiently generate draft tokens. The conventional approach (Leviathan et al., 2023; Chen et al., 2023) decomposes the next step generation into three steps:

- $\mathcal{M}^{(s)}$ proposes an unverified draft sequence with length $L$ by auto-regressive decoding.
- $\mathcal{M}^{(l)}$ evaluates posterior probabilities of $L$ draft tokens in parallel.
- $\tau$ tokens that retain the target distribution are accepted by a modified rejection sampling schema based on the draft sequence and the distribution gap.

Leviathan et al. (2023) demonstrate that the wall-clock time improvement ratio is directly proportional to $\tau$, while the arithmetic operation increment ratio is inversely proportional to $\tau$. Consequently, $\tau$, also known as the acceptance length, plays a crucial role in determining the performance of acceleration. This analysis also applies when using multiple draft sequences (Miao et al., 2024; Li et al., 2024c;b; Sun et al., 2024). Note that $\tau$ is closely related to the distribution gap between the target LLM and the draft model. With efficient decoding requirements, the draft model typically has limited capacity, resulting in a significant distribution gap compared to the target LLM. Fortunately, during inference, the acceptance rate is primarily influenced by the alignment of distributions on the desired tokens, i.e., the tokens to which the target LLM assigns high probabilities. However, previous speculative sampling works mainly focus on the entire vocabulary set w.r.t. knowledge distillation from the target LLM (Li et al., 2024b; Zhou et al., 2024), thereby disconnecting the training process from the practical decoding requirements.

**EAGLE** (Li et al., 2024b) is a lightweight draft model design, as shown in Figure 2. During decoding, it utilizes the LM Head of the target LLM to generate draft tokens. Specifically, we assume that the speculation starts from timestep $t$, meaning the first draft token is at timestep $t + 1$. To generate the draft token $x_{t+1}$, the target LLM's hidden state $f_{t-1}^{(l)}$ in the second-to-top layer is concatenated with the embedding $e_t$ to perform the input of the draft model. During training, EAGLE constructs a regression task between $f^{(l)}$s and the predicted hidden states $f^{(s)}$s of the draft model. However, due to the auto-regressive decoding, the draft model only accesses the target LLM's features at the beginning of the speculation. It uses the features produced by itself as input for subsequent steps. This context misalignment, stemming from feature inaccuracies, leads to error accumulation and hinders the performance of generating later draft tokens (Li et al., 2024b; Du et al., 2024). EAGLE-2 (Li et al., 2024c) employs the same model design but works on dynamic drafting structures instead of a static tree structure during the decoding stage, yet the aforementioned issue remains unresolved.

## 3 METHODOLOGY

As outlined before, previous speculative sampling methods suffer from disharmonies between training and decoding. This section introduces HArmonized Speculative Sampling (HASS) to tackle objective misalignment and context inconsistency through harmonized objective distillation and harmonized context alignment, respectively, as described below.

### 3.1 HARMONIZED OBJECTIVE DISTILLATION

HASS prioritizes the most decoding-desired tokens by leveraging the ranking distillation (Tang & Wang, 2018) idea from the recommender system. Specifically, ranking distillation aims to train a student model to assign higher ranks to the items that are top-ranked by the teacher model. In the context of speculative sampling, the draft model and the target LLM serve as the student and the teacher, respectively. Draft models with similar properties will perform at a higher acceptance rate in the decoding stage. Consider the set of K tokens with the highest probabilities from the target LLM's probability distribution as $\hat{\Omega} \subset \Omega$, where $\Omega$ represents the entire vocabulary. HASS considers the following Top-K distillation loss:

$$L_{\text{Top-K}} = -\sum_{x \in \hat{\Omega}} q(x) \log p(x), \qquad (1)$$

where $q$ and $p$ are the next token probability distributions of the target LLM and the draft model, respectively. Note that, when integrated with EAGLE, the training stage can obtain $\hat{\Omega}$ from hidden states of the target LLM. This implies that the proposed loss function benefits from the same efficient

training cost as EAGLE. We evaluate the proposed Top-K distillation loss against six alternative losses, such as BiLD (Li et al., 2024a) and Recall@k Surrogate loss (Patel et al., 2022), through ablation studies.

## 3.2 HARMONIZED CONTEXT ALIGNMENT

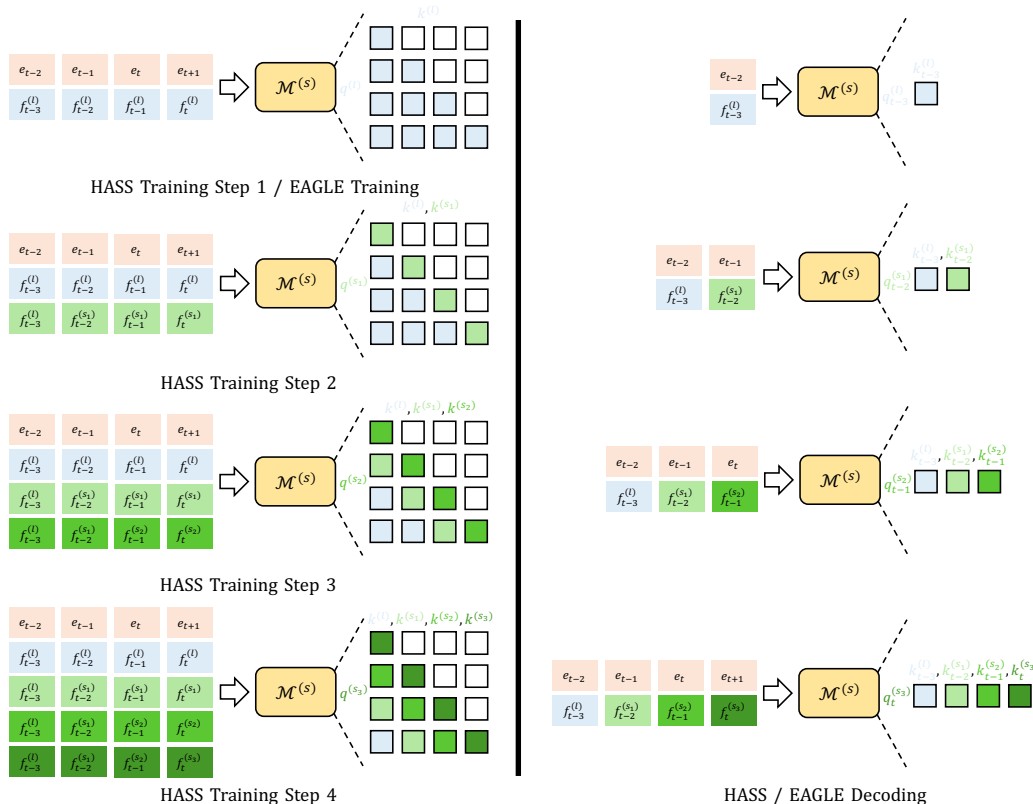

Figure 3: Training with harmonized context alignment, where $q$ and $k$ refer to the query and key states in the transformer layer, respectively. Superscript $(l)$ denotes tensors from the target LLM, and superscript $(s_j)$ denotes tensors from the $j$-th draft model forward. Note that during training $(s_j)$ refers to calling $j$ times draft model in a batch, while during inference $(s_j)$ refers to $j$-th auto-regressive decoding.

HASS follows a context alignment schema that aligns training and decoding on their contexts. The training procedure of HASS is divided into $n$ steps, enabling the draft model to utilize contextual features consistent with those in the decoding stage and addressing the context inconsistency by adapting the inaccurate features generated in previous HASS training steps. Specifically, it is achieved by first taking the inaccurate feature from the last draft model as query, and then considering the inaccuracy accumulation in key-value part of the transformer block.

Formally, in the HASS training step $j$, given input token sequence $x_1, x_2, .., x_T$, we optimize the draft model $\mathcal{M}^{(s)}$ with the following objective function:

$$\min_{\mathcal{M}^{(s)}} \sum_{t=1}^{T-1} [\text{CrossEntropy}(P^{(l)}(x_{t+1}|x_1, \ldots, x_t), P^{(s)}(x_{t+1}|x_1, \ldots, x_t)) + \text{Aux-loss}], \text{ where}$$

$$P^{(s)}(x_{t+1}|x_1, \ldots, x_t) = \text{Head}(f_{t+1}^{(s_j)})$$

$$= \text{Head}(\mathcal{M}^{(s)}(\underbrace{f_t^{(s_{j-1})}}_{\text{query}}, \underbrace{f_1^{(l)} \oplus \cdots \oplus f_{t-j+1}^{(l)} \oplus f_{t-j+2}^{(s_1)} \oplus \cdots \oplus f_t^{(s_{j-1})}}_{\text{key \& value}})),$$

where the labels above the braces read: "From last draft" (over $f_t^{(s_{j-1})}$), "From target LLM" (over $f_1^{(l)} \oplus \cdots \oplus f_{t-j+1}^{(l)}$), "From previous draft models" (over $f_{t-j+2}^{(s_1)} \oplus \cdots \oplus f_t^{(s_{j-1})}$).

| | | Temperature = 0 | | | | Temperature = 1 | | | |
|---|---|---|---|---|---|---|---|---|---|
| Model | Method | MT-bench | HumanEval | GSM8K | Mean | MT-bench | HumanEval | GSM8K | Mean |
| | PLD | 1.43 | 1.59 | 1.37 | 1.46 | - | - | - | - |
| | Lookahead | 1.66 | 1.77 | 1.65 | 1.69 | - | - | - | - |
| | SpS (V-68M) | 2.02 | 2.03 | 2.04 | 2.03 | 1.72 | 1.50 | 1.65 | 1.62 |
| | SpS (L-68M) | 1.83 | 1.81 | 1.83 | 1.82 | 1.47 | 1.36 | 1.46 | 1.43 |
| L2 7B | Medusa | 2.34 | 2.48 | 2.37 | 2.40 | 2.35 | 2.56 | 2.40 | 2.44 |
| | EAGLE | 3.68 | 3.90 | 3.77 | 3.78 | 3.45 | 3.67 | 3.62 | 3.58 |
| | EAGLE-2 | 4.44 | 4.78 | 4.60 | 4.61 | 4.23 | 4.47 | 4.50 | 4.40 |
| | **HASS** | **4.99** | **5.29** | **5.17** | **5.15** | **4.84** | **4.91** | **5.01** | **4.92** |
| | PLD | 1.46 | 1.70 | 1.44 | 1.53 | - | - | - | - |
| | Lookahead | 1.64 | 1.85 | 1.69 | 1.73 | - | - | - | - |
| | SpS (V-68M) | 2.13 | 2.61 | 2.21 | 2.32 | 1.73 | 2.25 | 1.81 | 1.93 |
| | SpS (L-68M) | 1.83 | 1.67 | 1.70 | 1.73 | 1.50 | 1.34 | 1.45 | 1.43 |
| L2 13B | Medusa | 2.51 | 2.56 | 2.70 | 2.59 | 2.53 | 2.89 | 2.72 | 2.71 |
| | EAGLE | 3.86 | 4.50 | 4.17 | 4.18 | 3.62 | 4.27 | 3.98 | 3.96 |
| | EAGLE-2 | 4.74 | 5.57 | 5.17 | 5.16 | 4.60 | 5.41 | 5.03 | 5.01 |
| | **HASS** | **5.13** | **6.05** | **5.55** | **5.58** | **4.98** | **5.86** | **5.41** | **5.42** |
| | EAGLE | 2.91 | 3.66 | 3.57 | 3.38 | 2.67 | 3.35 | 3.30 | 3.11 |
| L3 8B | EAGLE-2 | 4.21 | 4.93 | 4.42 | 4.52 | 3.90 | 4.73 | 4.30 | 4.31 |
| | **HASS** | **4.68** | **5.54** | **5.02** | **5.08** | **4.26** | **5.30** | **4.85** | **4.80** |
| | EAGLE | 3.24 | 4.07 | 3.79 | 3.70 | 3.06 | 3.85 | 3.66 | 3.52 |
| L3 70B | EAGLE-2 | 4.10 | 5.02 | 4.37 | 4.50 | 4.00 | 4.93 | 4.35 | 4.43 |
| | **HASS** | **4.62** | **5.78** | **5.24** | **5.21** | **4.59** | **5.68** | **5.20** | **5.16** |

Table 1: Acceptance lengths $\tau$ of different methods on MT-bench, HumanEval, and GSM8K datasets with temperature $T \in \{0, 1\}$. L2 represents LLaMA2-Chat, while L3 represents LLaMA3-Instruct. SpS stands for Vanilla Speculative Sampling, while V-68M and L-68M represent Vicuna-68M and LLaMA-68M, which are the draft models of SpS.

$P^{(l)}$ is the auto-regressive probability distribution provided by the target LLM, the Aux-loss consists of the proposed Top-K loss and the feature regression loss (following EAGLE), Head and $\oplus$ stand for the language modeling head and the concatenation operation respectively. When training tokens in the entire sequence in parallel, the above formulation adapts the inaccurate features from previous $j - 1$ steps for all positions except the first $j - 1$ positions. Note that compared to EAGLE, HASS takes additional training overhead due to the extra $n - 1$ training steps for adapting inaccurate features, while maintaining the same decoding overhead. To accelerate the training procedure, we propose a modification to the attention mask mechanism, as outlined below:

- The first step mirrors the training stage of EAGLE. At timestep $t + 1$, the draft model takes the target model's feature $f_t^{(l)}$ as input and produces the draft feature $f_{t+1}^{(s_1)}$. In this step, the attention mask remains the same as the original causal mask without any modification.

- In the second step, features from the first step are incorporated. For instance, in the self-attention mechanism at timestep $t + 1$, $f_t^{(s_1)}$ is used to calculate the current query. Keys and values are derived from $f_{:t}^{(l)} \oplus f_t^{(s_1)}$, where $f_{:t}^{(l)}$ includes features from timesteps earlier than $t$. The attention mask is adjusted to ensure that the previous feature seen by $f_i^{(s_1)}$ is always $f_{i-1}^{(l)}$, as shown in the 'HASS Training Step 2' part of Figure 3.

- For step $j \geq 3$, the feature from the previous step $f_t^{(s_{j-1})}$ is utilized to calculate the query at timestep $t + 1$, while keys and values are generated by $f_{:t-j+2}^{(l)} \oplus f_{t-j+2}^{(s_1)} \oplus \ldots \oplus f_t^{(s_{j-1})}$.

We empirically demonstrate that the acceleration effect converges with a small $n$ so that the training of HASS is cost-efficient. The actual training overhead of HASS in terms of training speed, computational cost, and GPU memory is investigated in Appendix A.8.

| Model | Method | Temperature = 0 | | | | Temperature = 1 | | | |
|---|---|---|---|---|---|---|---|---|---|
| | | MT-bench | HumanEval | GSM8K | Mean | MT-bench | HumanEval | GSM8K | Mean |
| L2 7B | SpS (V-68M) | 1.35x | 1.38x | 1.37x | 1.37x | 1.17x | 1.02x | 1.12x | 1.10x |
| | SpS (L-68M) | 1.23x | 1.24x | 1.25x | 1.24x | 1.00x | 0.94x | 0.99x | 0.98x |
| | Medusa | 1.91x | 1.96x | 2.20x | 2.02x | 2.00x | 2.25x | 2.15x | 2.13x |
| | EAGLE | 1.90x | 2.10x | 2.04x | 2.01x | 1.50x | 1.91x | 1.87x | 1.76x |
| | EAGLE-2 | 2.66x | 3.06x | 2.72x | 2.81x | 2.39x | 2.87x | 2.54x | 2.60x |
| | HASS | **2.99x** | **3.41x** | **3.32x** | **3.24x** | **2.70x** | **3.13x** | **2.87x** | **2.90x** |
| L2 13B | SpS (V-68M) | 1.63x | 1.98x | 1.68x | 1.76x | 1.33x | 1.72x | 1.39x | 1.48x |
| | SpS (L-68M) | 1.41x | 1.29x | 1.30x | 1.33x | 1.12x | 1.04x | 1.11x | 1.09x |
| | Medusa | 2.26x | 2.25x | 2.71x | 2.41x | 2.31x | 2.47x | 2.36x | 2.38x |
| | EAGLE | 1.80x | 2.46x | 2.41x | 2.22x | 1.84x | 2.10x | 2.21x | 2.05x |
| | EAGLE-2 | 3.02x | 3.64x | 3.23x | 3.30x | 3.04x | 3.45x | 3.13x | 3.21x |
| | HASS | **3.23x** | **4.24x** | **3.48x** | **3.65x** | **3.28x** | **3.78x** | **3.37x** | **3.48x** |
| L3 8B | EAGLE | 1.29x | 2.00x | 1.85x | 1.71x | 1.25x | 1.41x | 1.67x | 1.44x |
| | EAGLE-2 | 2.64x | 3.31x | 2.54x | 2.83x | 2.39x | 2.54x | 2.48x | 2.47x |
| | HASS | **2.78x** | **3.43x** | **3.06x** | **3.09x** | **2.49x** | **3.05x** | **2.89x** | **2.81x** |
| L3 70B | EAGLE | 2.14x | 2.74x | 2.42x | 2.43x | 1.80x | 2.34x | 2.12x | 2.09x |
| | EAGLE-2 | 2.94x | 3.98x | 3.19x | 3.37x | 3.02x | 3.61x | 3.21x | 3.28x |
| | HASS | **3.40x** | **4.68x** | **4.08x** | **4.05x** | **3.43x** | **4.25x** | **3.87x** | **3.85x** |

Table 2: Speedup ratios of different methods on MT-bench, HumanEval, and GSM8K datasets with temperature T ∈ {0, 1}. L2 represents LLaMA2-Chat, while L3 represents LLaMA3-Instruct. SpS stands for Vanilla Speculative Sampling, while V-68M and L-68M represent Vicuna-68M and LLaMA-68M, which are the draft models of SpS.

## 4  EXPERIMENT

### 4.1  EXPERIMENTAL SETUP

**Target LLMs.** LLaMA2-Chat 7/13B and LLaMA3-Instruct 8/70B.

**Tasks.** We conduct evaluations on three generation tasks. For multi-turn conversation, code generation, and mathematical reasoning tasks, we choose the MT-bench (Zheng et al., 2024), HumanEval (Chen et al., 2021), and GSM8K (Cobbe et al., 2021) datasets, respectively. The batch size is set as 1 under all the experiments following Leviathan et al. (2023) and Zhou et al. (2024).

**Metrics.** HASS neither fine-tunes the target LLMs' weights during training nor relaxes the acceptance conditions during decoding, making it a lossless acceleration method. Thus, the generation quality is promised with no need for evaluation. We use the following two metrics to measure the acceleration performance:

- **Speedup Ratio**: The actual test speedup ratio relative to vanilla auto-regressive decoding.

- **Acceptance Length** $\tau$: The average number of tokens generated per drafting-verification cycle, indicating the number of tokens accepted by the target LLM from the draft model.

Note that the speedup ratio is sensitive to the hardware due to variations in computing power, and the acceptance length may be slightly affected by hardware due to numerical errors. Therefore, all inference processes are conducted on NVIDIA H800 GPU.

**Comparisons.** The vanilla auto-regressive decoding is taken as the baseline, which serves as the benchmark for speedup ratios (1.00x). We compare HASS with recent lossless speculative sampling methods, including PLD (Saxena, 2023), Lookahead (Fu et al., 2023), Vanilla Speculative Sampling (Chen et al., 2023), Medusa (Cai et al., 2024), EAGLE (Li et al., 2024b), and EAGLE-2 (Li et al., 2024c). PLD and Lookahead are free of traning, which respectively use string matched from the prompt and cached n-grams as draft tokens instead of generating draft tokens from a draft model's predicted probability distribution. Therefore, the results of PLD and Lookahead under temperature = 1 are not reported in Table 1.

**Implementation.** Our code is built based on EAGLE-2's open-source repository[1]. Experiments on EAGLE and EAGLE-2 reuse draft model weights trained by Li et al. (2024b). For harmonized objective distillation, K is set as 10, and the loss of harmonized objective distillation is added to EAGLE's original loss with a coefficient of $w = 1.0$. For harmonized context alignment, the draft model is aligned for 3 steps during training. For dynamic tree structure, we set the total number of draft tokens to 60 for all experiments with a draft tree depth of 6. We keep other settings, such as the fixed training dataset, i.e., the ShareGPT[2] dataset with 68,000 dialogues, and the optimizer, consistent with EAGLE-2.

## 4.2 Effectiveness & Ablation Study

In this section, we first evaluate the effectiveness of HASS by comparing it with existing speculative sampling methods on acceptance length and speedup ratio. Then, we conduct ablation studies on harmonized objective distillation and harmonized context alignment. Inspired by Yi et al. (2024), we further conduct experiments by training on different proportions of the ShareGPT dataset to investigate HASS's scalability in the face of data sparsity (see Appendix A.6), and by evaluating on the translation tasks to investigate HASS's robustness across different task types (see Appendix A.7). As shown from the results, HASS is more scalable than EAGLE-2 with fewer training data and achieves promising improvements over EAGLE-2 on translation tasks consistent with results on MT-bench, HumanEval, and GSM8K.

### 4.2.1 Effectiveness

We present different methods' acceptance lengths and speedup ratios across three datasets in Tables 1 and 2, respectively. HASS performs the largest acceptance length and highest speedup ratio across all datasets and LLMs we tested. Most methods achieve their best performance on the HumanEval dataset, as the fixed templates in the code generation task are easier to draft and accelerate. Though PLD and Lookahead are free of training, they consistently show poorer performance than Medusa, EAGLE, EAGLE-2, and HASS.

### 4.2.2 Ablation Study on Harmonized Objective Distillation

We first study the effects of different K and the weight $w$ of the Top-K loss by varying these hyper-parameters and summarize the results in Figure 4. Training with the Top-K loss ($w > 0$) always improves performance compared to training without the Top-K loss ($w = 0$). HASS achieves the largest acceptance length when $w = 0.5$. A small value of K may result in performance degeneration, as the draft model only focuses on the token with the highest probability and consequently neglects other potential tokens. With a larger K, the Top-K loss generally brings better results, while the acceptance length is the largest when K = 5.

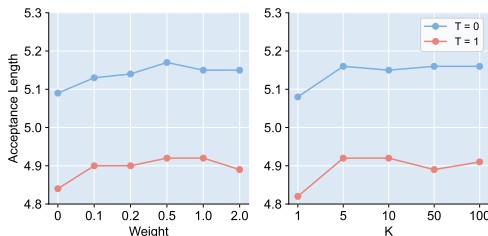

Figure 4: Acceptance lengths $\tau$ of HASS with varied Ks and weights of the Top-K loss. The results are conducted on LLaMA2-Chat 7B and averaged over MT-bench, HumanEval, and GSM8K datasets with temperature $T \in \{0, 1\}$.

Since the harmonized objective distillation can be implemented with any loss function that focuses on the most probable tokens w.r.t. the target distribution, we further consider the following loss functions and compare them with the Top-K Loss:

- Top-P Loss, where the $\hat{\Omega}$ is formed by the most probable tokens whose cumulative probability is just larger than P.

- Normed Top-K Loss, where the target and draft distributions are both normalized over $\hat{\Omega}$. The normalization operation can be either linear or softmax.

---

[1] https://github.com/SafeAILab/EAGLE
[2] https://huggingface.co/datasets/Aeala/ShareGPT_Vicuna_unfiltered

| Loss Function | Temperature = 0 | Temperature = 1 | Mean |
|---|---|---|---|
| Top-K Loss | 4.99 | **4.84** | **4.92** |
| Top-P Loss | 5.03 | 4.76 | 4.90 |
| Normed Top-K Loss (Linear) | 4.97 | 4.83 | 4.90 |
| Normed Top-K Loss (Softmax) | 4.98 | 4.72 | 4.85 |
| Bi-directional Top-K Loss | 4.99 | 4.72 | 4.86 |
| Recall@k Surrogate Loss | 4.97 | 4.76 | 4.87 |
| BiLD Loss | **5.04** | 4.75 | 4.90 |

Table 3: Acceptance lengths $\tau$ of HASS with different kinds of loss functions for harmonized objective distillation. The results are conducted on LLaMA2-Chat 7B over the MT-bench dataset with temperature T $\in \{0, 1\}$.

- Bi-directional Top-K Loss, where the distillation is conducted over the most probable tokens w.r.t. the target distribution as well as the draft distribution.
- Recall@k Surrogate Loss (Patel et al., 2022), where a smooth approximation of the recall metric is obtained and is differentiable for direct optimization.
- BiLD Loss (Li et al., 2024a), where the internal logits ranking information is captured by constructing logits differences with long-tail noise filtered out.

After searching the optimal hyper-parameters for each of the compared loss functions, we summarize their best results in Table 3. BiLD loss outperforms other loss functions under temperature T = 0, while Top-K loss outperforms others under temperature T = 1. Generally, Top-K loss shows the best performance. A better loss function may exist than Top-K loss to exploit the target LLM further. We leave this topic in future works.

We also conduct an experiment with LLaMA2-Chat 7B, where the fixed training dataset is replaced by the dataset generated by the target LLM (see Appendix A.4). We observe that when using non-greedy decoding, the acceptance length increases from 4.92 to 5.19 averaging over three datasets. Therefore, information obtained from harmonized objective distillation is not equivalent to direct distillation from target-model-generated data.

### 4.2.3 ABLATION STUDY ON HARMONIZED CONTEXT ALIGNMENT

| | Aligning Step | MT-bench | HumanEval | GSM8K | Mean |
|---|---|---|---|---|---|
| | EAGLE-2 + Top-K | 4.59 | 4.97 | 4.77 | 4.78 |
| | HASS Align-2 | 4.95 | 5.25 | 5.12 | 5.11 |
| T=0 | HASS Align-3 | **4.99** | 5.29 | 5.17 | 5.15 |
| | HASS Align-4 | **4.99** | **5.30** | **5.18** | **5.16** |
| | HASS Align-5 | 4.98 | 5.26 | 5.09 | 5.11 |
| | EAGLE-2 + Top-K | 4.46 | 4.61 | 4.64 | 4.57 |
| | HASS Align-2 | 4.71 | 4.89 | 4.98 | 4.86 |
| T=1 | HASS Align-3 | **4.84** | 4.91 | 5.01 | **4.92** |
| | HASS Align-4 | 4.77 | **4.93** | **5.03** | 4.91 |
| | HASS Align-5 | 4.71 | 4.92 | 4.95 | 4.86 |

Table 4: Acceptance lengths $\tau$ of HASS with varied aligning steps in the harmonized context alignment. The results are conducted on LLaMA2-Chat 7B with temperature T $\in \{0, 1\}$.

We propose the harmonized context alignment, which eliminates the feature inconsistency of draft models between the training and decoding stages. To study the effect of increasing the aligning steps in the harmonized context alignment, we conduct experiments by varying the step number and summarize the results in Table 4.

As the first training step of HASS is the same as EAGLE-2, we continually train EAGLE-2's draft model weights with the Top-K loss and consider it the baseline. Without harmonized context alignment (EAGLE-2 + Top-K), the draft model performs the worst across all datasets. Training with 3/4

steps of harmonized context alignment generally obtains the most considerable acceptance length. When training with 5 steps of context alignment, the acceptance length decreases. We believe this is caused by the draft model's limited capacity, as it predicts less accurately on former steps' tokens when paying too much attention to the latter ones. Figure 5 shows the acceptance rate $\alpha$ across speculation steps on the MT-bench dataset following Li et al. (2024c). In later speculation steps, HASS performs better acceptance rates than EAGLE-2, demonstrating the effectiveness of harmonized context alignment.

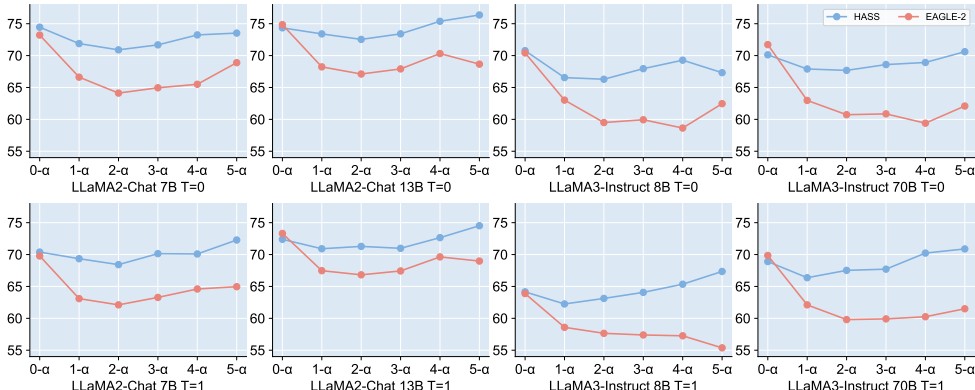

Figure 5: Acceptance rates $\alpha$ (%) of HASS and EAGLE-2 across different speculation steps on the MT-bench dataset with temperature $T \in \{0, 1\}$.

As shown in Figure 5, the acceptance rates of HASS decrease compared to EAGLE-2 on LLaMA2-Chat 13B and LLaMA3-Instruct 70B at the first step (0-$\alpha$). The draft models degrade on the first speculation step with much attention paid to the latter speculation steps, while the first step's acceptance rates are crucial to larger acceptance lengths. We conduct experiments to emphasize the significance of former speculation steps by reweighting the training loss from each step with a factor $\beta$. In specific, the step $j$'s training loss will be multiplied by $\beta^{j-1}$. Table 5 and Figure 6 show the acceptance lengths and acceptance rates of HASS with different reweight factors on LLaMA3-Instruct 70B over the MT-bench dataset, respectively. With the factor $\beta$ decreasing from 1.0 to 0.5, HASS achieves better acceptance lengths with different temperatures. Correspondingly, we perceive that the acceptance rate at the first speculation step is consistently higher with a smaller $\beta$, while the acceptance rates at the latter speculation steps generally decline. When the factor $\beta$ decreases to 0.3, too much emphasis is assigned to the first speculation step, leading to deterioration in acceptance length. Since the further exploration of an appropriate trade-off between different speculation steps is out of this paper's scope, we leave it for future work.

| Reweight Factor $\beta$ | T = 0 | T = 1 | Mean |
|---|---|---|---|
| 1.0 (Default) | 4.62 | 4.59 | 4.61 |
| 0.7 | 4.65 | 4.61 | 4.63 |
| 0.5 | **4.67** | **4.62** | **4.65** |
| 0.3 | 4.65 | 4.61 | 4.63 |

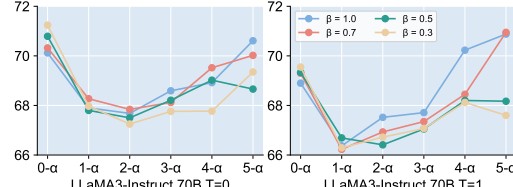

Table 5: Acceptance lengths $\tau$ of HASS with different reweight factors $\beta$ for harmonized context alignment. The results are conducted on LLaMA3-Instruct 70B over the MT-bench dataset with temperature $T \in \{0, 1\}$.

Figure 6: Acceptance rates $\alpha$ (%) of HASS with different reweight factors $\beta$ for harmonized context alignment. The results are conducted on LLaMA3-Instruct 70B over the MT-bench dataset with temperature $T \in \{0, 1\}$.

## 5 RELATED WORK

There have been a number of works on improving the acceptance rate of speculative sampling while maintaining the target distribution. Most of them fall into two categories. (1) The former category

is aligned training that tries to obtain draft models aligned with the target LLM before the decoding stage. Zhou et al. (2024) propose a knowledge distillation approach and study several strategies to improve the alignment. Li et al. (2024b) demonstrate that hidden states of the target LLM as input of the draft model provide extra feature uncertainty information. Xiao et al. (2024) also utilize hidden states of the target LLM and introduce an RNN-based draft model design that achieves a comparable acceptance rate. GLIDE (Du et al., 2024) instead reuses the KV cache of the target LLM. It also notices the context misalignment when using information from the target LLM, but the proposed blockwise attention mask method can not solve the misalignment completely. (2) The latter category is efficient decoding, which designs sophisticated decoding strategies to utilize concurrency efficiently. Miao et al. (2024) propose to utilize multiple draft models and design a tree-based attention mechanism to verify multiple draft sequences efficiently. Li et al. (2024c) introduce a dynamic structure to save computation by pruning inefficient paths in the draft tree. Sun et al. (2024) study improving the verification stage through optimal transportation. However, these works tend to only consider training or decoding, ignoring the linkage of these two stages. This work instead aims to link training and decoding, leading to harmonized speculative sampling.

## 6 CONCLUSION

This paper introduces HASS, a harmonized speculative sampling solution that addresses disharmonies between training and decoding on their objectives and contexts. Compared to its closest baseline, EAGLE-2, HASS improves the acceptance rate without any inference overhead. Experiments conducted on LLaMA2-Chat 7/13B and LLaMA3-Instruct 8/70B demonstrate the effectiveness and efficiency of HASS. Averaging on MT-bench, HumanEval, and GSM8K, HASS is 2.81x-4.05x faster than vanilla auto-regressive decoding, 8%-20% faster than EAGLE-2.

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

# A APPENDIX

## A.1 IMPLEMENTATION OF HARMONIZED CONTEXT ALIGNMENT

We present the pseudo code of harmonized context alignment, which is implemented without the customized attention mask, for better understanding. The actual implementation in our experiments is achieved by the customized attention mask as shown in Figure 3.

```python
def train_batch(
        draft_model,              # draft model
        lm_head,                  # language model head
        optimizer,                # optimizer
        forward_num,              # aligning steps in harmonized context alignment
        hidden_states_target,     # target LLM's feature
        input_ids,                # input tokens
    ):
    hidden_states_draft_list = []
    for forward_idx in range(forward_num):
        optimizer.zero_grad()
        predict = draft_model(hidden_states_target, input_ids, hidden_states_draft_list)
        hidden_states_draft = torch.cat([hidden_states_target[:, :1], predict[:, :-1]], dim=1).detach()
        hidden_states_draft_list.append(hidden_states_draft)
        target_head, pred_head = lm_head(hidden_states_target), lm_head(predict)
        loss = feature_loss(hidden_states_target, predict) + logit_loss(target_head, pred_head)
        loss.backward()
        optimizer.step()
```

```python
def attention(
        hidden_states_target,        # target LLM's feature
        attention_mask,              # causal attention mask
        hidden_states_draft_list,    # list of draft model's features
    ):
    bs, seq_len = hidden_states_target.shape[0], hidden_states_target.shape[1]
    query = q_proj(hidden_states_draft_list[-1]) if hidden_states_draft_list else q_proj(hidden_states_target)
    key_t, value_t = k_proj(hidden_states_target), v_proj(hidden_states_target)
    attn_weight = torch.matmul(query, key_t.transpose(2, 3)) / math.sqrt(query.shape[-1]) + attention_mask
    indices = torch.arange(seq_len)
    for i, hidden_states_draft in enumerate(hidden_states_draft_list[::-1]):
        key_d, ind_q, ind_k = k_proj(hidden_states_draft), indices[i:], indices[:seq_len - i]
        attn_weight_d = torch.matmul(query, key_d.transpose(2, 3)) / math.sqrt(query.shape[-1])
        attn_weight[:, :, ind_q, ind_k] = attn_weight_d[:, :, ind_q, ind_k]
    attn_weight_normed = F.softmax(attn_weight, dim=-1)
    attn_output = torch.matmul(attn_weight_normed, value_t)
    for i, hidden_states_draft in enumerate(hidden_states_draft_list[::-1]):
        value_d, ind_q, ind_k = v_proj(hidden_states_draft), indices[i:], indices[:seq_len - i]
        attn_output[:, :, ind_q] += attn_weight[:, :, ind_q, ind_k][..., None] * (value_d[:, :, ind_k] -
     value_t[:, :, ind_k])
    attn_output = o_proj(attn_output.transpose(1, 2).reshape(bs, seq_len, -1))
    return attn_output
```

## A.2 HARMONIZED CONTEXT ALIGNMENT ON TOKENS

In this section, we attempt to verify whether applying token alignment as well as feature alignment brings better performance. In specific, we use the tokens generated by the draft model for training in harmonized context alignment instead of using the tokens from training data. We apply feature and token alignment to EAGLE-2's draft model weights and summarize the results in Table 6 and Figure 7.

|  | Temperature = 0 | Temperature = 1 | Mean |
|---|---|---|---|
| EAGLE-2 | 4.44 | 4.23 | 4.34 |
| Feature Only | **4.83** | **4.60** | **4.72** |
| Feature + Token (0.1) | 4.81 | 4.57 | 4.69 |
| Feature + Token (0.2) | 4.78 | 4.51 | 4.65 |
| Feature + Token (1.0) | 4.28 | 4.11 | 4.20 |

Table 6: Acceptance lengths $\tau$ of applying feature and token alignment to EAGLE-2's draft model weights, where 'Token $(x)$' denotes tokens from training data being replaced by draft-model-generated tokens with a probability of $x$. The results are conducted on LLaMA2-Chat 7B over the MT-bench dataset with temperature $T \in \{0, 1\}$.

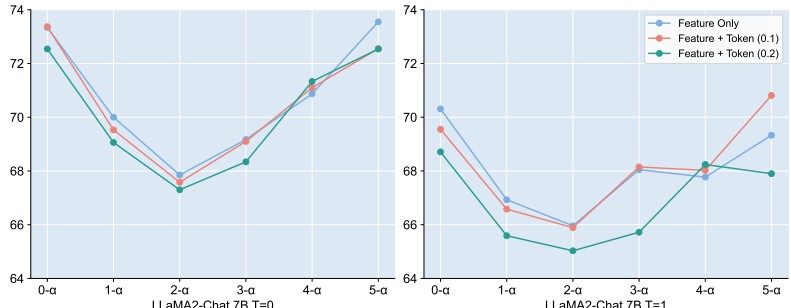

Figure 7: Acceptance rates $\alpha$ (%) of applying feature and token alignment to EAGLE-2's draft model weights, where 'Token $(x)$' denotes tokens from training data being replaced by draft-model-generated tokens with a probability of $x$. The results are conducted on LLaMA2-Chat 7B over the MT-bench dataset with temperature $T \in \{0, 1\}$.

Feature only alignment brings the best performance, while adding token alignment leads to degeneration. With the probability of applying token alignment increasing from 0.1 to 1.0, the acceptance length decreases consistently. As shown in Figure 7, more token alignment generally causes lower acceptance rates. As a result, training with tokens generated by the draft model in harmonized context alignment hurts the acceleration performance.

## A.3 HYPER-PARAMETERS OF TOP-K LOSS

We conduct an ablation study on Top-K loss's hyper-parameters, i.e., K and $w$, in Section 4.2.2 and show the averaged acceptance lengths over three datasets in Figure 4. Here, we present the speedup ratios and acceptance lengths of HASS with varied Ks and $w$s in Table 7.

| | K | $w$ | MT-bench Speedup | $\tau$ | HumanEval Speedup | $\tau$ | GSM8K Speedup | $\tau$ | Mean Speedup | $\tau$ |
|---|---|---|---|---|---|---|---|---|---|---|
| T=0 | 1 | 1.0 | 2.89x | 4.94 | 3.24x | 5.19 | 3.11x | 5.10 | 3.08x | 5.08 |
| | 5 | 1.0 | 2.90x | 5.00 | 3.44x | 5.29 | 3.33x | **5.18** | 3.22x | 5.16 |
| | 10 | 1.0 | 2.99x | 4.99 | 3.41x | 5.29 | 3.32x | 5.17 | 3.24x | 5.15 |
| | 50 | 1.0 | 2.85x | 5.01 | **3.46x** | 5.29 | 3.41x | 5.17 | 3.24x | 5.16 |
| | 100 | 1.0 | 2.93x | 5.00 | 3.45x | 5.29 | 3.45x | **5.18** | 3.28x | 5.16 |
| | 10 | 0.0 | 2.77x | 4.93 | 3.38x | 5.22 | 3.18x | 5.11 | 3.11x | 5.09 |
| | 10 | 0.1 | 2.98x | 4.96 | 3.40x | 5.26 | **3.51x** | 5.16 | **3.30x** | 5.13 |
| | 10 | 0.2 | 2.87x | 4.98 | 3.41x | 5.29 | 3.35x | 5.16 | 3.21x | 5.14 |
| | 10 | 0.5 | **3.00x** | **5.02** | 3.32x | **5.31** | 3.50x | **5.18** | 3.27x | **5.17** |
| | 10 | 2.0 | 2.94x | 4.98 | 3.37x | 5.29 | 3.34x | 5.17 | 3.22x | 5.15 |
| T=1 | 1 | 1.0 | 2.58x | 4.70 | 2.79x | 4.80 | 2.83x | 4.95 | 2.73x | 4.82 |
| | 5 | 1.0 | 2.64x | 4.81 | 3.13x | 4.94 | 2.93x | 5.02 | 2.90x | **4.92** |
| | 10 | 1.0 | **2.70x** | **4.84** | 3.13x | 4.91 | 2.87x | 5.01 | 2.90x | **4.92** |
| | 50 | 1.0 | 2.62x | 4.77 | 3.01x | 4.88 | **2.99x** | **5.03** | 2.87x | 4.89 |
| | 100 | 1.0 | 2.66x | 4.74 | 3.14x | **4.97** | 2.90x | **5.03** | 2.90x | 4.91 |
| | 10 | 0.0 | 2.61x | 4.71 | 2.76x | 4.84 | 2.79x | 4.96 | 2.72x | 4.84 |
| | 10 | 0.1 | 2.69x | 4.75 | 3.05x | 4.94 | 2.87x | 5.00 | 2.87x | 4.90 |
| | 10 | 0.2 | 2.66x | 4.75 | **3.16x** | 4.95 | 2.88x | 5.01 | 2.90x | 4.90 |
| | 10 | 0.5 | 2.68x | 4.80 | 3.15x | 4.93 | 2.96x | **5.03** | **2.93x** | **4.92** |
| | 10 | 2.0 | 2.68x | 4.75 | 3.11x | 4.89 | 2.89x | **5.03** | 2.89x | 4.89 |

Table 7: Speedup ratios and acceptance lengths $\tau$ of HASS with varied Ks and $w$s of the Top-K loss on LLaMA2-Chat 7B over MT-bench, HumanEval, and GSM8K datasets with temperature T $\in \{0, 1\}$.

## A.4 SELF-DISTILLATION

In the main text, we use the fixed ShareGPT dataset to train draft models for a fair comparison with EAGLE and EAGLE-2. Following existing speculative sampling methods (Zhou et al., 2024; Cai et al., 2024), we further use target-model-generated outputs to distill the draft model from the target model's real output distribution, dubbed as self-distillation. In specific, we feed the prompts from the ShareGPT dataset into the target models recursively with temperature set to 0 and collect the responses as multi-turn conversations for self-distillation following Li et al. (2024b). To study the effect of self-distillation, we conduct experiments on HASS and EAGLE-2 by training the draft model with fixed data or model-generated data and summarize the results in Table 8.

| | | | | MT-bench | | HumanEval | | GSM8K | | Mean | |
|---|---|---|---|---|---|---|---|---|---|---|---|
| | Model | Method | Data | Speedup | $\tau$ | Speedup | $\tau$ | Speedup | $\tau$ | Speedup | $\tau$ |
| T=0 | L2 7B | EAGLE-2 | F | 2.66x | 4.44 | 3.06x | 4.78 | 2.72x | 4.60 | 2.81x | 4.61 |
| | | | MG | **2.86x** | **4.70** | **3.30x** | **5.12** | **3.03x** | **5.00** | **3.06x**(+0.25) | **4.94**(+0.33) |
| | | HASS | F | 2.99x | 4.99 | 3.41x | 5.29 | 3.32x | 5.17 | 3.24x | 5.15 |
| | | | MG | **3.13x** | **5.25** | **3.85x** | **5.70** | **3.40x** | **5.57** | **3.46x**(+0.22) | **5.51**(+0.36) |
| | L2 13B | EAGLE-2 | F | 3.02x | 4.74 | **3.64x** | **5.57** | **3.23x** | **5.17** | **3.30x** | **5.16** |
| | | | MG | **3.04x** | **4.80** | 3.47x | 5.46 | 3.19x | 5.16 | 3.23x(-0.07) | 5.14(-0.02) |
| | | HASS | F | 3.23x | 5.13 | 4.24x | **6.05** | 3.48x | 5.55 | 3.65x | 5.58 |
| | | | MG | **3.34x** | **5.27** | **4.42x** | 6.00 | **3.63x** | **5.61** | **3.80x**(+0.15) | **5.63**(+0.05) |
| T=1 | L2 7B | EAGLE-2 | F | 2.39x | 4.23 | 2.87x | 4.47 | 2.54x | 4.50 | 2.60x | 4.40 |
| | | | MG | **2.49x** | **4.38** | **2.94x** | **4.73** | **2.69x** | **4.80** | **2.71x**(+0.11) | **4.64**(+0.24) |
| | | HASS | F | 2.70x | 4.84 | 3.13x | 4.91 | 2.87x | 5.01 | 2.90x | 4.92 |
| | | | MG | **2.75x** | **4.97** | **3.39x** | **5.24** | **3.13x** | **5.35** | **3.09x**(+0.19) | **5.19**(+0.27) |
| | L2 13B | EAGLE-2 | F | 3.04x | 4.60 | **3.45x** | **5.41** | **3.13x** | **5.03** | **3.21x** | **5.01** |
| | | | MG | **3.08x** | **4.63** | 3.23x | 5.25 | 3.04x | 4.95 | 3.12x(-0.09) | 4.94(-0.07) |
| | | HASS | F | 3.28x | 4.98 | **3.78x** | **5.86** | 3.37x | 5.41 | 3.48x | **5.42** |
| | | | MG | **3.33x** | **5.02** | 3.76x | 5.80 | **3.60x** | 5.42 | **3.56x**(+0.08) | 5.41(-0.01) |

Table 8: Speedup ratios and acceptance lengths $\tau$ of HASS and EAGLE-2 with fixed or target-model-generated training data. F and MG stand for 'Fixed' and 'Model-Generated', respectively. L2 represents LLaMA2-Chat.

On LLaMA2-Chat 7B, self-distillation consistently brings improvements for HASS and EAGLE-2. On LLaMA2-Chat 13B, self-distillation only achieves marginally better or comparable results, which is consistent with the observation from Li et al. (2024b) ('data from the target LLM marginally improves performance' in its section 4.3.3). Especially, the acceptance lengths of the self-distilled HASS are lower than that of the vanilla HASS on the HumanEval dataset, while both the speedup ratios and acceptance lengths of the self-distilled EAGLE-2 are lower than that of the vanilla EAGLE-2 on HumanEval and GSM8K datasets. It may be due to the code generation dataset HumanEval and the mathematical reasoning dataset GSM8K being less similar to the training dataset ShareGPT compared with MT-bench.

HASS outperforms EAGLE-2 on either fixed training data or model-generated training data. It is noted that HASS trained on the fixed dataset even achieves better performance than EAGLE-2 trained on the model-generated data consistently. With self-distillation, HASS consistently achieves more improvement or less degeneration in terms of the acceptance length compared with EAGLE-2.

## A.5 DRAFTING HYPER-PARAMETERS

Li et al. (2024c) find that the draft token's confidence score is strongly positively correlated with the acceptance rate, and accordingly propose the context-aligned dynamic draft tree, which can be dynamically adjusted with two hyper-parameters: 'depth' and 'number of tokens'. 'Depth' decides the draft tree's depth during the expansion phase, while 'number of tokens' decides how many draft tokens will be kept during the reranking phase. Increasing both these hyper-parameters surely leads to a larger acceptance length. Nevertheless, sending more draft tokens into the target model for verification causes a higher overhead in real applications. Therefore, we vary these hyper-parameters and report the speedup ratios in Table 9 to find a better trade-off.

| | Depth | | 5 | | | | 6 | | | | 7 | | | | 8 | | | | 9 | | |
|---|---|---|---|---|---|---|---|---|---|---|---|---|---|---|---|---|---|---|---|---|---|
| | # Tokens | 40 | 60 | 80 | 100 | 40 | 60 | 80 | 100 | 40 | 60 | 80 | 100 | 40 | 60 | 80 | 100 | 40 | 60 | 80 | 100 |
| T=0 L2 7B | EAGLE-2 | 2.48x | 2.78x | 2.61x | 2.69x | 2.69x | 2.66x | 2.70x | 2.79x | 2.71x | 2.86x | 2.91x | **2.95x** | 2.60x | 2.60x | 2.88x | 2.89x | 2.28x | 2.54x | 2.55x | 2.65x |
| | HASS-MG | 3.09x | 3.02x | 3.04x | 3.22x | 3.08x | 3.13x | 3.19x | 3.22x | 3.14x | 3.11x | 3.29x | 3.27x | 3.07x | 3.16x | 3.31x | **3.32x** | 2.78x | 2.75x | 2.95x | 3.03x |
| L2 13B | EAGLE-2 | 2.63x | 2.96x | 3.04x | 3.06x | 3.01x | 3.02x | 3.18x | 3.22x | 2.78x | 3.12x | 3.14x | 3.24x | 2.98x | 3.12x | 3.19x | **3.26x** | 2.49x | 2.64x | 2.69x | 2.72x |
| | HASS-MG | 3.25x | 3.31x | 3.24x | 3.25x | 3.33x | 3.34x | **3.49x** | 3.40x | 3.19x | 3.42x | 3.36x | 3.40x | 3.15x | 3.40x | 3.40x | 3.37x | 2.70x | 2.74x | 3.09x | 3.02x |
| T=1 L2 7B | EAGLE-2 | 2.31x | 2.37x | 2.55x | 2.36x | 2.42x | 2.39x | 2.33x | 2.40x | 2.49x | **2.66x** | 2.65x | 2.44x | 2.36x | 2.48x | 2.38x | 2.64x | 2.29x | 2.22x | 2.27x | 2.42x |
| | HASS-MG | 2.79x | 2.89x | 2.86x | 2.88x | 2.72x | 2.75x | **2.92x** | 2.82x | 2.83x | 2.76x | 2.81x | 2.75x | 2.49x | 2.68x | 2.77x | 2.77x | 2.30x | 2.35x | 2.58x | 2.50x |
| L2 13B | EAGLE-2 | 2.92x | 3.11x | 2.88x | 2.79x | 3.06x | 3.04x | **3.16x** | 2.93x | 3.05x | 3.14x | 3.14x | 3.11x | 3.00x | 3.13x | 3.15x | 2.98x | 2.61x | 2.72x | 2.65x | 2.54x |
| | HASS-MG | 3.24x | 3.30x | 3.27x | 3.19x | 3.33x | 3.33x | **3.40x** | 3.28x | 3.19x | 3.26x | 3.24x | 3.26x | 3.15x | 3.26x | 3.19x | 3.17x | 2.62x | 2.77x | 2.74x | 2.84x |

Table 9: Speedup ratios of EAGLE-2 and HASS-MG with varied depths and numbers of tokens on the MT-bench dataset with temperature $T \in \{0, 1\}$, where HASS-MG denotes HASS trained with self-distillation. L2 represents LLaMA2-Chat.

When 'depth' $= 5$, the acceptance length is relatively small. When 'depth' $= 9$, the verification overhead is extremely high. Thus, neither of these settings achieves a promising speedup ratio. For both HASS-MG and EAGLE-2, the best performances are achieved when 'depth' $\in \{6, 7, 8\}$ and '# tokens' $\in \{60, 80, 100\}$. HASS-MG consistently obtains a superior performance compared with EAGLE-2 through hyper-parameter tuning across different LLMs and temperatures.

## A.6 NUMBER OF TRAINING TOKENS

Inspired by Yi et al. (2024), we randomly sample different proportions of the training dataset, i.e., the ShareGPT dataset with 68,000 dialogues, to investigate the influences of training token numbers. In specific, we train the draft models of HASS and EAGLE-2 with $1/8$, $1/4$, $1/2$ and the entire ShareGPT dataset and summarize the results in Figure 8 and Table 10.

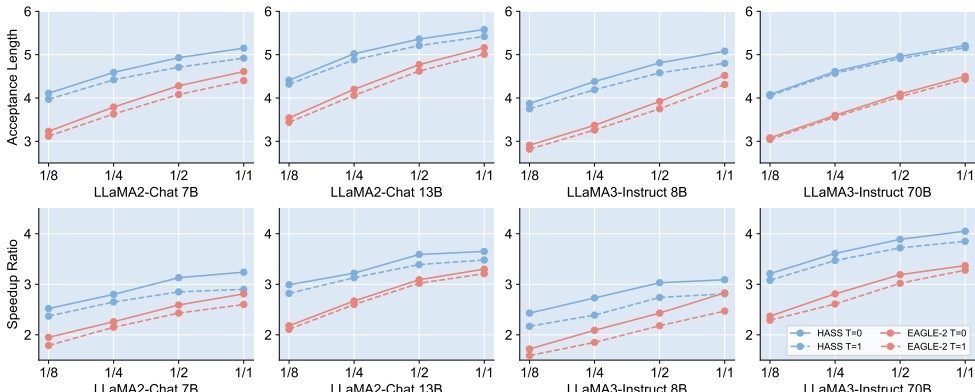

Figure 8: Acceptance lengths $\tau$ and speedup ratios of HASS and EAGLE-2 averaging across MT-bench, HumanEval, and GSM8K with different proportions of training dataset, i.e., the ShareGPT dataset with 68,000 dialogues.

As shown from Figure 8, HASS consistently outperforms EAGLE-2 under different proportions of training dataset with temperature $T \in \{0, 1\}$. HASS with merely $1/4$ training dataset achieves better or comparable performance compared to EAGLE-2 with the entire training dataset, which demonstrates HASS's superior data exploitation and scalability obtained through further aligning on objectives and contexts between training and decoding. The speedup ratio and acceptance length of HASS and EAGLE-2 are approximately logarithmically proportional to the scale of training data, which is consistent with the finding in Yi et al. (2024). As shown from Table 10, the decrease in training data contributes to more severe degradation on EAGLE-2 than that on HASS, reflecting HASS's robustness to data sparsity.

| | Model | Method | Proportion | MT-bench | | HumanEval | | GSM8K | | Mean | |
|---|---|---|---|---|---|---|---|---|---|---|---|
| | | | | Speedup | $\tau$ | Speedup | $\tau$ | Speedup | $\tau$ | Speedup | $\tau$ |
| T=0 | L2 7B | EAGLE-2 | 1/8 | 1.74x | 3.06 | 2.00x | 3.39 | 2.12x | 3.24 | 1.95x | 3.23 |
| | | | 1/4 | 2.08x | 3.64 | 2.49x | 3.93 | 2.21x | 3.81 | 2.26x | 3.79 |
| | | | 1/2 | 2.36x | 4.11 | 2.76x | 4.46 | 2.64x | 4.28 | 2.59x | 4.28 |
| | | | 1/1 | 2.66x | 4.44 | 3.06x | 4.78 | 2.72x | 4.60 | 2.81x | 4.61 |
| | | HASS | 1/8 | 2.32x | 3.92 | 2.69x | 4.30 | 2.56x | 4.12 | 2.52x | 4.11 |
| | | | 1/4 | 2.64x | 4.42 | 3.05x | 4.76 | 2.70x | 4.59 | 2.80x | 4.59 |
| | | | 1/2 | 2.85x | 4.79 | 3.36x | 5.05 | 3.18x | 4.94 | 3.13x | 4.93 |
| | | | 1/1 | 2.99x | 4.99 | 3.41x | 5.29 | 3.32x | 5.17 | 3.24x | 5.15 |
| | L2 13B | EAGLE-2 | 1/8 | 1.88x | 3.25 | 2.39x | 3.79 | 2.27x | 3.57 | 2.18x | 3.54 |
| | | | 1/4 | 2.38x | 3.82 | 2.85x | 4.53 | 2.79x | 4.25 | 2.67x | 4.20 |
| | | | 1/2 | 2.74x | 4.32 | 3.46x | 5.18 | 3.08x | 4.81 | 3.09x | 4.77 |
| | | | 1/1 | 3.02x | 4.74 | 3.64x | 5.57 | 3.23x | 5.17 | 3.30x | 5.16 |
| | | HASS | 1/8 | 2.59x | 4.01 | 3.37x | 4.79 | 3.01x | 4.43 | 2.99x | 4.41 |
| | | | 1/4 | 2.89x | 4.55 | 3.59x | 5.51 | 3.18x | 5.00 | 3.22x | 5.02 |
| | | | 1/2 | 3.16x | 4.90 | 4.20x | 5.86 | 3.41x | 5.31 | 3.59x | 5.36 |
| | | | 1/1 | 3.23x | 5.13 | 4.24x | 6.05 | 3.48x | 5.55 | 3.65x | 5.58 |
| | L3 8B | EAGLE-2 | 1/8 | 1.54x | 2.77 | 2.00x | 3.12 | 1.61x | 2.83 | 1.72x | 2.91 |
| | | | 1/4 | 1.89x | 3.18 | 2.30x | 3.59 | 2.08x | 3.35 | 2.09x | 3.37 |
| | | | 1/2 | 2.24x | 3.68 | 2.64x | 4.16 | 2.42x | 3.91 | 2.43x | 3.92 |
| | | | 1/1 | 2.64x | 4.21 | 3.31x | 4.93 | 2.54x | 4.42 | 2.83x | 4.52 |
| | | HASS | 1/8 | 2.14x | 3.61 | 2.84x | 4.22 | 2.30x | 3.78 | 2.43x | 3.87 |
| | | | 1/4 | 2.46x | 4.04 | 3.24x | 4.83 | 2.48x | 4.27 | 2.73x | 4.38 |
| | | | 1/2 | 2.72x | 4.43 | 3.38x | 5.28 | 2.99x | 4.71 | 3.03x | 4.81 |
| | | | 1/1 | 2.78x | 4.68 | 3.43x | 5.54 | 3.06x | 5.02 | 3.09x | 5.08 |
| | L3 70B | EAGLE-2 | 1/8 | 2.09x | 2.87 | 2.72x | 3.45 | 2.29x | 2.92 | 2.37x | 3.08 |
| | | | 1/4 | 2.47x | 3.33 | 3.25x | 4.01 | 2.71x | 3.46 | 2.81x | 3.60 |
| | | | 1/2 | 2.76x | 3.74 | 3.71x | 4.57 | 3.11x | 3.96 | 3.19x | 4.09 |
| | | | 1/1 | 2.94x | 4.10 | 3.98x | 5.02 | 3.19x | 4.37 | 3.37x | 4.50 |
| | | HASS | 1/8 | 2.73x | 3.68 | 3.79x | 4.61 | 3.10x | 3.95 | 3.21x | 4.08 |
| | | | 1/4 | 3.05x | 4.12 | 4.23x | 5.18 | 3.56x | 4.54 | 3.61x | 4.61 |
| | | | 1/2 | 3.27x | 4.40 | 4.52x | 5.57 | 3.87x | 4.92 | 3.89x | 4.96 |
| | | | 1/1 | 3.40x | 4.62 | 4.68x | 5.78 | 4.08x | 5.24 | 4.05x | 5.21 |
| T=1 | L2 7B | EAGLE-2 | 1/8 | 1.60x | 2.99 | 1.90x | 3.21 | 1.86x | 3.17 | 1.79x | 3.12 |
| | | | 1/4 | 1.89x | 3.48 | 2.28x | 3.71 | 2.27x | 3.71 | 2.15x | 3.63 |
| | | | 1/2 | 2.26x | 3.93 | 2.57x | 4.14 | 2.46x | 4.17 | 2.43x | 4.08 |
| | | | 1/1 | 2.39x | 4.23 | 2.87x | 4.47 | 2.54x | 4.50 | 2.60x | 4.40 |
| | | HASS | 1/8 | 2.19x | 3.82 | 2.50x | 4.06 | 2.41x | 4.03 | 2.37x | 3.97 |
| | | | 1/4 | 2.50x | 4.27 | 2.84x | 4.46 | 2.61x | 4.52 | 2.65x | 4.42 |
| | | | 1/2 | 2.63x | 4.56 | 3.10x | 4.75 | 2.81x | 4.81 | 2.85x | 4.71 |
| | | | 1/1 | 2.70x | 4.84 | 3.13x | 4.91 | 2.87x | 5.01 | 2.90x | 4.92 |
| | L2 13B | EAGLE-2 | 1/8 | 1.91x | 3.16 | 2.18x | 3.71 | 2.23x | 3.46 | 2.11x | 3.44 |
| | | | 1/4 | 2.31x | 3.68 | 2.72x | 4.40 | 2.77x | 4.11 | 2.60x | 4.06 |
| | | | 1/2 | 2.78x | 4.20 | 3.19x | 5.00 | 3.08x | 4.67 | 3.02x | 4.62 |
| | | | 1/1 | 3.04x | 4.60 | 3.45x | 5.41 | 3.13x | 5.03 | 3.21x | 5.01 |
| | | HASS | 1/8 | 2.49x | 3.94 | 2.98x | 4.70 | 2.99x | 4.33 | 2.82x | 4.32 |
| | | | 1/4 | 2.87x | 4.43 | 3.40x | 5.35 | 3.11x | 4.87 | 3.13x | 4.88 |
| | | | 1/2 | 3.22x | 4.75 | 3.70x | 5.69 | 3.26x | 5.18 | 3.39x | 5.21 |
| | | | 1/1 | 3.28x | 4.98 | 3.78x | 5.86 | 3.37x | 5.41 | 3.48x | 5.42 |
| | L3 8B | EAGLE-2 | 1/8 | 1.51x | 2.64 | 1.63x | 3.01 | 1.64x | 2.81 | 1.59x | 2.82 |
| | | | 1/4 | 1.77x | 2.99 | 1.87x | 3.51 | 1.92x | 3.29 | 1.85x | 3.26 |
| | | | 1/2 | 1.90x | 3.40 | 2.25x | 4.03 | 2.38x | 3.82 | 2.18x | 3.75 |
| | | | 1/1 | 2.39x | 3.90 | 2.54x | 4.73 | 2.48x | 4.30 | 2.47x | 4.31 |
| | | HASS | 1/8 | 1.96x | 3.42 | 2.31x | 4.10 | 2.25x | 3.72 | 2.17x | 3.75 |
| | | | 1/4 | 2.22x | 3.77 | 2.51x | 4.63 | 2.45x | 4.18 | 2.39x | 4.19 |
| | | | 1/2 | 2.43x | 4.10 | 2.96x | 5.07 | 2.82x | 4.56 | 2.74x | 4.58 |
| | | | 1/1 | 2.49x | 4.26 | 3.05x | 5.30 | 2.89x | 4.85 | 2.81x | 4.80 |
| | L3 70B | EAGLE-2 | 1/8 | 2.16x | 2.85 | 2.52x | 3.35 | 2.19x | 2.91 | 2.29x | 3.04 |
| | | | 1/4 | 2.26x | 3.29 | 3.01x | 3.94 | 2.57x | 3.44 | 2.61x | 3.56 |
| | | | 1/2 | 2.70x | 3.67 | 3.37x | 4.47 | 3.00x | 3.94 | 3.02x | 4.03 |
| | | | 1/1 | 3.02x | 4.00 | 3.61x | 4.93 | 3.21x | 4.35 | 3.28x | 4.43 |
| | | HASS | 1/8 | 2.80x | 3.70 | 3.46x | 4.52 | 2.98x | 3.93 | 3.08x | 4.05 |
| | | | 1/4 | 3.10x | 4.10 | 3.90x | 5.11 | 3.41x | 4.51 | 3.47x | 4.57 |
| | | | 1/2 | 3.28x | 4.36 | 4.15x | 5.46 | 3.72x | 4.91 | 3.72x | 4.91 |
| | | | 1/1 | 3.43x | 4.59 | 4.25x | 5.68 | 3.87x | 5.20 | 3.85x | 5.16 |

Table 10: Speedup ratios and acceptance lengths $\tau$ of HASS and EAGLE-2 with different proportions of training dataset, i.e., the ShareGPT dataset with 68,000 dialogues. L2 represents LLaMA2-Chat, while L3 represents LLaMA3-Instruct.

## A.7 EVALUATION ON TRANSLATION TASKS

To investigate the robustness of HASS across different task types, we further evaluate HASS and EAGLE-2 on five translation tasks[3] by following Yi et al. (2024). It is noted that both HASS and EAGLE-2 are trained on the fixed ShareGPT dataset without adaptation for translation tasks. We conduct experiments on LLaMA2-Chat 7/13B and LLaMA3-Instruct 8/70B and summarize the results in Table 11.

| | Model | Method | De→En Speedup | $\tau$ | Fr→En Speedup | $\tau$ | Ja→En Speedup | $\tau$ | Ru→En Speedup | $\tau$ | Zh→En Speedup | $\tau$ | Mean Speedup | $\tau$ |
|---|---|---|---|---|---|---|---|---|---|---|---|---|---|---|
| T=0 | L2 7B | EAGLE-2 | 2.58x | 4.06 | 2.47x | 3.99 | 2.46x | 3.79 | 2.23x | 3.48 | 2.39x | 3.68 | 2.43x | 3.80 |
| | | HASS | **3.15x** | **4.55** | **2.99x** | **4.60** | **2.97x** | **4.26** | **2.64x** | **3.82** | **2.83x** | **4.10** | **2.92x** | **4.27** |
| | L2 13B | EAGLE-2 | 2.95x | 4.51 | 3.00x | 4.41 | 2.67x | 3.80 | 2.61x | 3.65 | 2.60x | 3.92 | 2.77x | 4.06 |
| | | HASS | **3.63x** | **5.01** | **3.64x** | **4.94** | **3.05x** | **4.07** | **3.06x** | **4.03** | **3.02x** | **4.22** | **3.28x** | **4.45** |
| | L3 8B | EAGLE-2 | 2.59x | 3.89 | 2.34x | 3.96 | 1.91x | 2.97 | 1.90x | 3.25 | 2.03x | 3.17 | 2.15x | 3.45 |
| | | HASS | **2.98x** | **4.30** | **2.79x** | **4.21** | **2.21x** | **3.19** | **2.28x** | **3.53** | **2.32x** | **3.38** | **2.52x** | **3.72** |
| | L3 70B | EAGLE-2 | 3.10x | 4.17 | 3.13x | 4.07 | 2.35x | 3.16 | 2.79x | 3.76 | 2.52x | 3.39 | 2.78x | 3.71 |
| | | HASS | **3.75x** | **4.71** | **3.61x** | **4.47** | **2.67x** | **3.41** | **3.39x** | **4.25** | **2.84x** | **3.72** | **3.25x** | **4.12** |
| T=1 | L2 7B | EAGLE-2 | 2.26x | 3.86 | 2.41x | 3.91 | 2.09x | 3.58 | 1.98x | 3.34 | 2.25x | 3.61 | 2.20x | 3.66 |
| | | HASS | **2.80x** | **4.44** | **2.99x** | **4.59** | **2.66x** | **4.11** | **2.53x** | **3.74** | **2.65x** | **4.05** | **2.73x** | **4.19** |
| | L2 13B | EAGLE-2 | 2.97x | 4.29 | 2.77x | 4.31 | 2.45x | 3.73 | 2.33x | 3.51 | 2.47x | 3.72 | 2.60x | 3.91 |
| | | HASS | **3.45x** | **4.88** | **3.22x** | **4.84** | **3.02x** | **4.13** | **2.79x** | **3.97** | **2.83x** | **4.01** | **3.06x** | **4.37** |
| | L3 8B | EAGLE-2 | 2.23x | 3.67 | 2.21x | 3.69 | 1.85x | 2.79 | 1.94x | 3.15 | 1.89x | 3.03 | 2.02x | 3.27 |
| | | HASS | **2.80x** | **4.13** | **2.73x** | **4.08** | **2.21x** | **3.18** | **2.35x** | **3.54** | **2.11x** | **3.34** | **2.44x** | **3.65** |
| | L3 70B | EAGLE-2 | 2.97x | 4.02 | 2.95x | 3.89 | 2.37x | 3.16 | 2.72x | 3.65 | 2.44x | 3.34 | 2.69x | 3.61 |
| | | HASS | **3.71x** | **4.67** | **3.49x** | **4.38** | **2.83x** | **3.47** | **3.25x** | **4.11** | **2.75x** | **3.70** | **3.21x** | **4.07** |

Table 11: Speedup ratios and acceptance lengths $\tau$ of HASS and EAGLE-2 on five translation tasks, where draft models are trained with the fixed ShareGPT dataset. De, Fr, Ja, Ru, Zh, and En stand for German, French, Japanese, Russian, Chinese, and English, respectively. L2 represents LLaMA2-Chat, while L3 represents LLaMA3-Instruct.

As shown from Table 11, HASS consistently outperforms EAGLE-2 under all settings. HASS achieves 2.44x-3.28x wall-clock time speedup ratio averaging across five translation tasks, surpassing EAGLE-2 by 17%-24%. In terms of acceptance length, HASS achieves 8%-14% improvement over EAGLE-2. In consistent with results on dialogue (MT-bench), code generation (HumanEval), and mathematical reasoning (GSM8K) tasks, HASS shows promising improvements over EAGLE-2 on translation tasks, reflecting its robustness across different task types.

---

[3]https://github.com/Kthyeon/Multilingual-SpecBench

## A.8 TRAINING OVERHEAD

As shown from Table 4, training with 3/4 steps of harmonized context alignment generally obtains the most considerable acceptance length, and so the aligning step of HASS is fixed to 3 (Standard) in this paper unless stated otherwise. To investigate the actual training overhead of HASS, we train draft models for LLaMA2-Chat 7/13B and LLaMA3-Instruct 8/70B on a single NVIDIA H800 GPU with batch size set to 2 and varied aligning steps, and summarize the results of training speed, computational cost, and GPU memory in Figures 9, 10, and 11, respectively. It is worth mentioning that the training overhead of HASS with 1 aligning step is the same as that of EAGLE-2.

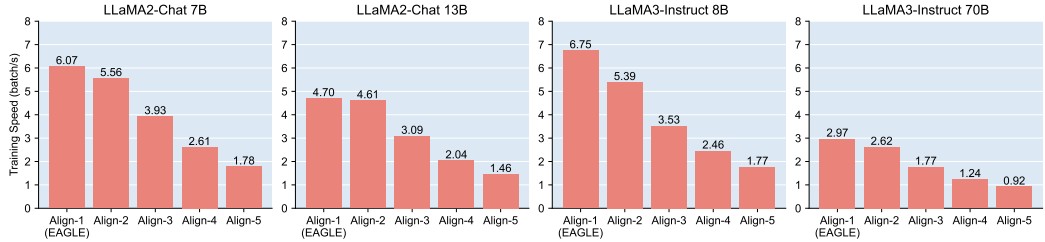

Figure 9: Training speed (batch/s) of HASS with varied aligning steps, where the speed of Align-1 is the same as that of EAGLE/EAGLE-2.

The training speed is evaluated by how many batches can be processed in one second, i.e., batch/s, and the ratio between Align-1 and Align-$j$ represents how much training time needed for executing the same amount of training data compared to EAGLE-2. As shown from Figure 9, the training speed decreases with more aligning steps. However, the actual training time of standard HASS (Align-3) is only 66.34% more than EAGLE-2 averaging over four target models, and the highest extra time cost compared to EAGLE-2 is just 91.47% (on LLaMA3-Instruct 8B). The training overhead of HASS is totally affordable, while HASS achieves superior performance and requires unchanged inference overhead.

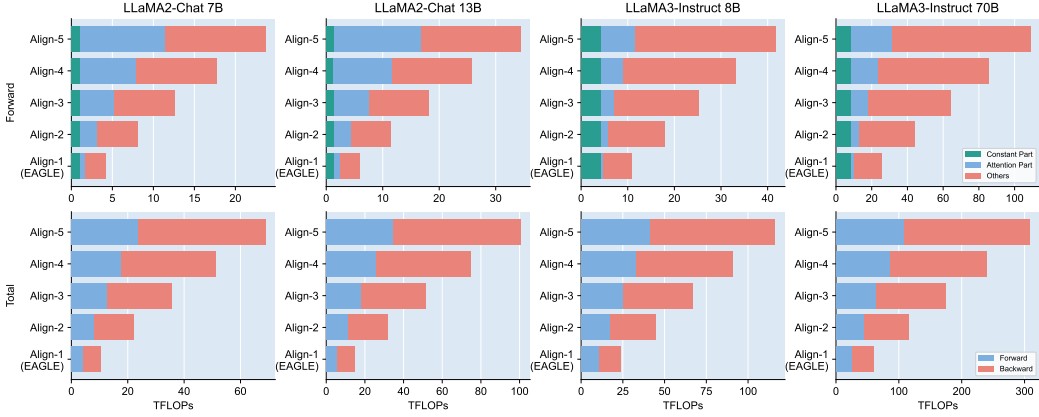

Figure 10: Training FLOPs of HASS with varied aligning steps, where the computational cost of Align-1 is the same as that of EAGLE/EAGLE-2. The upper figures show the FLOPs of the forward pass, while the lower figures show the FLOPs in total (forward and backward passes).

The computational cost is evaluated by TFLOPs and can be divided into forward pass and backward pass, we depict the cost of forward and backward passes in upper and lower figures in Figure 10, respectively. The cost of forward pass is consisted of three parts:

- Constant part is invariant to the number of aligning steps. Mapping target LLM's hidden state into $q(x)$ (refer to section 3.1) with the LM head for distilling the draft model is included in constant part.

- Attention part is linearly proportional to the hidden state number fed into the draft model, which is accumulated across HASS training steps, i.e., $\sum_{i=1}^{j} i$ for Align-$j$. Fusing token embeddings with hidden states, projecting hidden states into keys and values, and conducting attention operations between query and several kv-pairs sourced from different hidden states are included in attention part.

- Others is linearly proportional to the number of aligning steps, i.e., $j$ for Align-$j$. Computational costs except for constant and attention parts are included in others.

The cost of backward pass can be considered as $(A + O) \times 2$, where $A$ and $O$ represent attention part and others respectively, as the computation of constant part requires no gradient. Generally, the standard HASS (Align-3) requires approximately 3x computational cost of EAGLE-2.

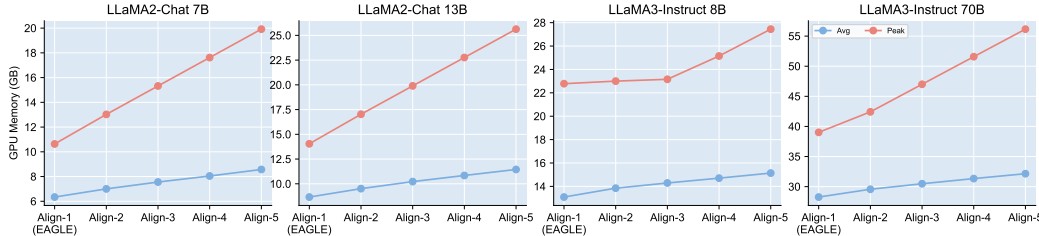

Figure 11: Training GPU memory of HASS with varied aligning steps, where the GPU memory of Align-1 is the same as that of EAGLE/EAGLE-2. Avg and Peak stand for the average and peak GPU memory across the training process, respectively.

The GPU memory is evaluated by GB and we report the average and peak GPU memory across the training process. Both the average and peak GPU memory increase with more aligning steps. The GPU memory requirement can be covered by a single NVIDIA H800 GPU even at Align-5 and batch size set to 2.

