# OpenReview forum: "Learning Harmonized Representations for Speculative Sampling"
_ICLR.cc/2025/Conference — ICLR 2025 Poster_

### Official Review · Reviewer_p2CZ · 2024-10-30

**Soundness:** 3
**Presentation:** 2
**Contribution:** 3
**Rating:** 6
**Confidence:** 4

**Summary:**

This paper proposes HASS, a speculative sampling algorithm. HASS makes two main improvements:

- Training objective. The loss focuses on the k most likely tokens in the target distribution, rather than fitting the complete target distribution.

- Training strategy. Methods like EAGLE have inconsistent distributions between training and inference phases - training uses hidden states from target mode, while inference involves predicted values. HASS simulates this process during training by using model predictions as training data.

The paper evaluated HASS on dialogue, code, and math tasks. On LLaMA models, HASS improved the acceptance length by 8%-16% compared to EAGLE-2, achieving 2.81-4.05x speedup.

**Strengths:**

- The research addresses LLM latency reduction, which is highly relevant for LLM applications.

- The experiments are comprehensive, covering various tasks and models, achieving 3-4x speedup compared to vanilla decoding.

- The ablation studies are thorough. They clearly demonstrate the impact of HASS's two improvements and how different parameter settings affect the final results.

**Weaknesses:**

- HASS builds upon EAGLE but requires higher training overhead compared to EAGLE.
- The paper lacks comparisons with more related methods, such as vanilla speculative sampling and Medusa.

Thank you for the author's response. The additional results are consistent with my expectations, so I have decided to maintain my previous score.

**Questions:**

- The authors state "It takes $n$ times the training overhead of EAGLE", suggesting HASS training steps have the same overhead as EAGLE training steps. However, HASS training steps (except step 0) actually have larger KV caches, attention scores, etc. For the $i$-th step, how much does the HASS training step increase compared to the EAGLE training step in terms of overhead (computational FLOPs, memory access bytes, GPU memory usage, actual execution time, and actual peak GPU memory)?

- The presentation could be clearer. In Figure 3, the light green text is difficult to read. In Table 1, the "-" marks for PLD and Lookahead Temperature=1 should be explained, considering readers may not be familiar with these methods.

---

> ### Author Response · Authors · 2024-11-20
> **Response to Reviewer p2CZ (1)**
>
> **Thanks for your valuable comments.**
>
> **Q1: "HASS builds upon EAGLE but requires higher training overhead compared to EAGLE." & "The authors state "It takes n times the training overhead of EAGLE", suggesting HASS training steps have the same overhead as EAGLE training steps. However, HASS training steps (except step 0) actually have larger KV caches, attention scores, etc. For the i-th step, how much does the HASS training step increase compared to the EAGLE training step in terms of overhead (computational FLOPs, memory access bytes, GPU memory usage, actual execution time, and actual peak GPU memory)?"**
>
> **A1-a:** It is worth mentioning that the harmonized context alignment's acceleration effect converges with 3/4 HASS training steps (refer to HASS's section 4.2.3) and all the experiments in our paper are conducted with 3 aligning steps unless stated otherwise. While the training overhead of HASS increases compared to EAGLE-2, it requires no extra inference overhead, and can be considered as a "free lunch" as mentioned by reviewer ETQd.
>
> To investigate the actual training overhead of HASS, we train draft models on a single NVIDIA H800 GPU with batch size set to 2 and varied aligning steps, and summarize the results of training speed, computational cost, and GPU memory in Appendix A.8 Training Overhead. It is noted that the training overhead of HASS with 1 alignment step is the same as that of EAGLE-2, and HASS's aligning step is always set to 3 (Standard) in all the experiments unless stated otherwise.
>
> We show the actual training speed (batch per second) and time ratio w.r.t. EAGLE-2 of HASS in the following table. The ratio between Align-1 and Align-$j$ represents how much training time needed for executing the same amount of training data compared to EAGLE-2.
>
> | Model               | Aligning Step      | Training Speed (batch/s) | Time Ratio w.r.t. EAGLE-2 |
> | ------------------- | ------------------ | ------------------------ | ------------------------- |
> | LLaMA2-Chat 7B      | Align-1 (EAGLE-2)  | 6.0687                   | 1.0000                    |
> |                     | Align-2            | 5.5562                   | 1.0922                    |
> |                     | Align-3 (Standard) | **3.9342**               | **1.5425**                |
> |                     | Align-4            | 2.6116                   | 2.3237                    |
> |                     | Align-5            | 1.7786                   | 3.4121                    |
> | LLaMA2-Chat 13B     | Align-1 (EAGLE-2)  | 4.7040                   | 1.0000                    |
> |                     | Align-2            | 4.6057                   | 1.0213                    |
> |                     | Align-3 (Standard) | **3.0872**               | **1.5237**                |
> |                     | Align-4            | 2.0352                   | 2.3113                    |
> |                     | Align-5            | 1.4568                   | 3.2290                    |
> | LLaMA3-Instruct 8B  | Align-1 (EAGLE-2)  | 6.7504                   | 1.0000                    |
> |                     | Align-2            | 5.3898                   | 1.2524                    |
> |                     | Align-3 (Standard) | **3.5256**               | **1.9147**                |
> |                     | Align-4            | 2.4573                   | 2.7471                    |
> |                     | Align-5            | 1.7698                   | 3.8142                    |
> | LLaMA3-Instruct 70B | Align-1 (EAGLE-2)  | 2.9678                   | 1.0000                    |
> |                     | Align-2            | 2.6205                   | 1.1325                    |
> |                     | Align-3 (Standard) | **1.7745**               | **1.6725**                |
> |                     | Align-4            | 1.2391                   | 2.3951                    |
> |                     | Align-5            | 0.9240                   | 3.2119                    |
>
> As shown from the table, the training speed decreases with more aligning steps. However, the actual training time of standard HASS (Align-3) is only 66.34% more than EAGLE-2 averaging over four target LLMs, and the highest extra time cost compared to EAGLE-2 is just 91.47% (on LLaMA3-Instruct 8B). The training overhead of HASS is totally affordable, while HASS achieves superior performance and requires unchanged inference overhead.

---

> ### Author Response · Authors · 2024-11-20
> **Response to Reviewer p2CZ (2)**
>
> **A1-b:** We show the computational cost (TFLOPs) for training HASS in the following table, and more detailed analyses can be found in Figure 10 Appendix A.8.
>
> | Aligning Step      | LLaMA2-Chat 7B | LLaMA2-Chat 13B | LLaMA3-Instruct 8B | LLaMA3-Instruct 70B |
> | ------------------ | -------------- | --------------- | ------------------ | ------------------- |
> | Align-1 (EAGLE-2)  | 10.5084 TFLOPs | 14.9715 TFLOPs  | 23.8149 TFLOPs     | 59.5859 TFLOPs      |
> | Align-2            | 22.0072 TFLOPs | 31.6964 TFLOPs  | 44.7723 TFLOPs     | 115.1055 TFLOPs     |
> | Align-3 (Standard) | 35.5701 TFLOPs | 51.5168 TFLOPs  | 67.1753 TFLOPs     | 175.1655 TFLOPs     |
> | Align-4            | 51.1970 TFLOPs | 74.4326 TFLOPs  | 91.0237 TFLOPs     | 239.7657 TFLOPs     |
> | Align-5            | 68.8880 TFLOPs | 100.4439 TFLOPs | 116.3178 TFLOPs    | 308.9064 TFLOPs     |
>
> As shown from the table, the standard HASS (Align-3) requires approximately 3x computational cost of EAGLE-2.
>
>
> **A1-c:** We show the average and peak GPU memory (GB) across the training process of HASS in the following table.
>
> |                    | LLaMA2-Chat 7B |           | LLaMA2-Chat 13B |           | LLaMA3-Instruct 8B |           | LLaMA3-Instruct 70B |           |
> | ------------------ | -------------- | --------- | --------------- | --------- | ------------------ | --------- | ------------------- | --------- |
> | Aligning Step      | Average (GB)   | Peak (GB) | Average (GB)    | Peak (GB) | Average (GB)       | Peak (GB) | Average (GB)        | Peak (GB) |
> | Align-1 (EAGLE-2)  | 6.3417         | 10.6349   | 8.6595          | 14.0431   | 13.0865            | 22.7815   | 28.2694             | 39.0179   |
> | Align-2            | 7.0027         | 13.0252   | 9.5080          | 17.0275   | 13.8451            | 23.0002   | 29.5636             | 42.4221   |
> | Align-3 (Standard) | 7.5551         | 15.3230   | 10.2210         | 19.8945   | 14.2904            | 23.1566   | 30.4787             | 46.9994   |
> | Align-4            | 8.0448         | 17.6187   | 10.8410         | 22.7618   | 14.7033            | 25.1484   | 31.3378             | 51.5791   |
> | Align-5            | 8.5708         | 19.9170   | 11.4482         | 25.6288   | 15.1372            | 27.4452   | 32.1469             | 56.1563   |
>
> As shown from the table, both the average and peak GPU memory increase with more aligning steps. The GPU memory requirement can be covered by a single NVIDIA H800 GPU even at Align-5 and batch size set to 2.

---

> ### Author Response · Authors · 2024-11-20
> **Response to Reviewer p2CZ (3)**
>
> **Q2: "The paper lacks comparisons with more related methods, such as vanilla speculative sampling and Medusa."**
>
> **A2:** We choose EAGLE-2 as one of our baselines, because it is considered as one of the fastest speculative sampling methods which consistently outperforms Vanilla Speculative Sampling and Medusa in EAGLE-2's original paper [A].
>
> Thanks for your valuable suggestion and we are working on the results of Vanilla Speculative Sampling and Medusa. We will update once the results are available.
>
> [A] Li et al., 2024. EAGLE-2: Faster Inference of Language Models with Dynamic Draft Trees, EMNLP 2024.

---

> ### Author Response · Authors · 2024-11-20
> **Response to Reviewer p2CZ (4)**
>
> **Q3: "The presentation could be clearer. In Figure 3, the light green text is difficult to read. In Table 1, the "-" marks for PLD and Lookahead Temperature=1 should be explained, considering readers may not be familiar with these methods."**
>
> **A3:** Thanks for your valuable suggestions.
>
> We will replace the colors in Figure 3 with more distinguishable ones.
>
> PLD and Lookahead are free of traning, which respectively use string matched from the prompt and cached n-grams as draft tokens instead of generating draft tokens from a draft model's predicted probability distribution. Therefore, the results of PLD and Lookahead under temperature=1 are not reported in Table 1.

---

> > ### Comment · Reviewer_p2CZ · 2024-11-24
> >
> > Thank you for your efforts in the response. The training overhead of HASS is acceptable.

---

> > > ### Author Response · Authors · 2024-11-25
> > > **Response to Reviewer p2CZ (5)**
> > >
> > > **Thanks for your reply.**
> > >
> > > We show the speedup ratios and acceptance lengths $\tau$ of Vanilla Speculative Sampling, Medusa and HASS across MT-bench, HumanEval, and GSM8K in the following table. SpS stands for Vanilla Speculative Sampling, while Vicuna-68M [A] (https://huggingface.co/double7/vicuna-68m) and LLaMA-68M [B] (https://huggingface.co/JackFram/llama-68m) represent the draft models of SpS.
> > >
> > > |             |                 |                  | MT-bench  |          | HumanEval |          | GSM8K     |          | Mean      |          |
> > > | ----------- | --------------- | ---------------- | --------- | -------- | --------- | -------- | --------- | -------- | --------- | -------- |
> > > | Temperature | Model           | Method           | Speedup   | $\tau$   | Speedup   | $\tau$   | Speedup   | $\tau$   | Speedup   | $\tau$   |
> > > | Temp=0      | LLaMA2-Chat 7B  | SpS (Vicuna-68M) | 1.35x     | 2.02     | 1.38x     | 2.03     | 1.37x     | 2.04     | 1.37x     | 2.03     |
> > > |             |                 | SpS (LLaMA-68M)  | 1.23x     | 1.83     | 1.24x     | 1.81     | 1.25x     | 1.83     | 1.24x     | 1.82     |
> > > |             |                 | Medusa           | 1.91x     | 2.34     | 1.96x     | 2.48     | 2.20x     | 2.37     | 2.02x     | 2.40     |
> > > |             |                 | HASS             | **2.99x** | **4.99** | **3.41x** | **5.29** | **3.32x** | **5.17** | **3.24x** | **5.15** |
> > > |             | LLaMA2-Chat 13B | SpS (Vicuna-68M) | 1.63x     | 2.13     | 1.98x     | 2.61     | 1.68x     | 2.21     | 1.76x     | 2.32     |
> > > |             |                 | SpS (LLaMA-68M)  | 1.41x     | 1.83     | 1.29x     | 1.67     | 1.30x     | 1.70     | 1.33x     | 1.73     |
> > > |             |                 | Medusa           | 2.26x     | 2.51     | 2.25x     | 2.56     | 2.71x     | 2.70     | 2.41x     | 2.59     |
> > > |             |                 | HASS             | **3.23x** | **5.13** | **4.24x** | **6.05** | **3.48x** | **5.55** | **3.65x** | **5.58** |
> > > | Temp=1      | LLaMA2-Chat 7B  | SpS (Vicuna-68M) | 1.17x     | 1.72     | 1.02x     | 1.50     | 1.12x     | 1.65     | 1.10x     | 1.62     |
> > > |             |                 | SpS (LLaMA-68M)  | 1.00x     | 1.47     | 0.94x     | 1.36     | 0.99x     | 1.46     | 0.98x     | 1.43     |
> > > |             |                 | Medusa           | 2.00x     | 2.35     | 2.25x     | 2.56     | 2.15x     | 2.40     | 2.13x     | 2.44     |
> > > |             |                 | HASS             | **2.70x** | **4.84** | **3.13x** | **4.91** | **2.87x** | **5.01** | **2.90x** | **4.92** |
> > > |             | LLaMA2-Chat 13B | SpS (Vicuna-68M) | 1.33x     | 1.73     | 1.72x     | 2.25     | 1.39x     | 1.81     | 1.48x     | 1.93     |
> > > |             |                 | SpS (LLaMA-68M)  | 1.12x     | 1.50     | 1.04x     | 1.34     | 1.11x     | 1.45     | 1.09x     | 1.43     |
> > > |             |                 | Medusa           | 2.31x     | 2.53     | 2.47x     | 2.89     | 2.36x     | 2.72     | 2.38x     | 2.71     |
> > > |             |                 | HASS             | **3.28x** | **4.98** | **3.78x** | **5.86** | **3.37x** | **5.41** | **3.48x** | **5.42** |
> > >
> > > As shown from the table, HASS consistently outperforms SpS (Vicuna-68M/LLaMA-68M) and Medusa under all settings, reflecting its superior performance achieved by harmonizing the draft model's training and decoding stages on their objectives and contexts.
> > >
> > > [A] Li et al., 2024. EAGLE-2: Faster Inference of Language Models with Dynamic Draft Trees, EMNLP 2024.
> > >
> > > [B] Miao et al., 2024. SpecInfer: Accelerating Large Language Model Serving with Tree-based Speculative Inference and Verification, ASPLOS 2024.

---

### Official Review · Reviewer_ETQd · 2024-11-03

**Soundness:** 3
**Presentation:** 3
**Contribution:** 3
**Rating:** 8
**Confidence:** 4

**Summary:**

This paper identifies two drawbacks—objective misalignment and context inconsistency—within the current speculative approach (EAGLE) and proposes a progressive training framework, HASS, for better adapting draft models to actual decoding scenarios. The framework incorporates a distillation loss and a progressive training paradigm to reduce error accumulation caused by misaligned input features, empowering draft models to generate longer and more reliable token sequences. Experiments across various datasets and model sizes demonstrate that HASS significantly outperforms other speculative competitors and achieves impressive inference acceleration.

**Strengths:**

1. The motivation to improve the current speculative decoding method is clear and well-founded.
2. The paper is well-written and easy to follow.
3. The proposed HASS significantly outperforms other baseline methods and achieves impressive speed-up performance. Without introducing extra inference costs, HASS can be nearly considered a 'free lunch' while incurring only acceptable additional training costs (3 or 4 extra alignment steps).
4. The authors conduct interesting ablation experiments to provide a better understanding of this method, such as the variations in reweight factors demonstrated in the experiments shown in Table 5.

**Weaknesses:**

1. The training efficiency of this method appears to have limitations. During the additional training steps (step 2, 3, 4...), each token’s corresponding KV matrix differs, making it infeasible to compute all attention scores through a single matrix multiplication. Could the authors offer a solution for this issue or provide detailed reports on the specific training time costs?
2. Section 3.2 (HARMONIZED CONTEXT ALIGNMENT) is the core of the proposed method, but the description is overly brief. To enhance readers' understanding, the authors could consider adding more technical details and method insights into the design choices. For instance, highlighting that different steps in model training focus on generating tokens for specific positions within the draft sequence would provide valuable context.

**Questions:**

1. What does the vertical axis of the line charts in Figures 5 and 6 represent? The ranges shown do not appear to correspond to the acceptance rate.
If the authors are willing to address our concerns, we are open to raise the scores during the rebuttal stage.

---

> ### Author Response · Authors · 2024-11-20
> **Response to Reviewer ETQd (1)**
>
> **Thanks for your valuable comments.**
>
> **Q1: "The training efficiency of this method appears to have limitations. During the additional training steps (step 2, 3, 4...), each token’s corresponding KV matrix differs, making it infeasible to compute all attention scores through a single matrix multiplication. Could the authors offer a solution for this issue or provide detailed reports on the specific training time costs?"**
>
> **A1:** In the attention operation, we add an extra dimension into the query and key matrices to adapt them to more than one piece of feature as input, i.e., the query and key matrices are shaped as (1, batch_size, num_head, seq_len, head_dim) and (feat_num, batch_size, num_head, seq_len, head_dim), respectively. Thus, we can compute all attention scores through a single matrix multiplication. You can refer to line 311 in https://github.com/HArmonizedSS/HASS/blob/main/model/cnets_hass.py for the real implementation.
>
> To investigate the training overhead of HASS, we train draft models on a single NVIDIA H800 GPU with batch size set to 2 and varied aligning steps, and summarize the results in Appendix A.8 Training Overhead. It is worth mentioning that the training overhead of HASS with 1 alignment step is the same as that of EAGLE-2, and HASS's aligning step is always set to 3 (Standard) in all the experiments unless stated otherwise.
>
> We show the actual training speed (batch per second) and time ratio w.r.t. EAGLE-2 of HASS in the following table. The ratio between Align-1 and Align-$j$ represents how much training time needed for executing the same amount of training data compared to EAGLE-2.
>
> | Model               | Aligning Step      | Training Speed (batch/s) | Time Ratio w.r.t. EAGLE-2 |
> | ------------------- | ------------------ | ------------------------ | ------------------------- |
> | LLaMA2-Chat 7B      | Align-1 (EAGLE-2)  | 6.0687                   | 1.0000                    |
> |                     | Align-2            | 5.5562                   | 1.0922                    |
> |                     | Align-3 (Standard) | **3.9342**               | **1.5425**                |
> |                     | Align-4            | 2.6116                   | 2.3237                    |
> |                     | Align-5            | 1.7786                   | 3.4121                    |
> | LLaMA2-Chat 13B     | Align-1 (EAGLE-2)  | 4.7040                   | 1.0000                    |
> |                     | Align-2            | 4.6057                   | 1.0213                    |
> |                     | Align-3 (Standard) | **3.0872**               | **1.5237**                |
> |                     | Align-4            | 2.0352                   | 2.3113                    |
> |                     | Align-5            | 1.4568                   | 3.2290                    |
> | LLaMA3-Instruct 8B  | Align-1 (EAGLE-2)  | 6.7504                   | 1.0000                    |
> |                     | Align-2            | 5.3898                   | 1.2524                    |
> |                     | Align-3 (Standard) | **3.5256**               | **1.9147**                |
> |                     | Align-4            | 2.4573                   | 2.7471                    |
> |                     | Align-5            | 1.7698                   | 3.8142                    |
> | LLaMA3-Instruct 70B | Align-1 (EAGLE-2)  | 2.9678                   | 1.0000                    |
> |                     | Align-2            | 2.6205                   | 1.1325                    |
> |                     | Align-3 (Standard) | **1.7745**               | **1.6725**                |
> |                     | Align-4            | 1.2391                   | 2.3951                    |
> |                     | Align-5            | 0.9240                   | 3.2119                    |
>
> As shown from the table, the training speed decreases with more aligning steps. However, the actual training time of standard HASS (Align-3) is only 66.34% more than EAGLE-2 averaging over four target LLMs, and the highest extra time cost compared to EAGLE-2 is just 91.47% (on LLaMA3-Instruct 8B). The training overhead of HASS is totally affordable, while HASS achieves superior performance and requires unchanged inference overhead.

---

> ### Author Response · Authors · 2024-11-20
> **Response to Reviewer ETQd (2)**
>
> **Q2: "Section 3.2 (HARMONIZED CONTEXT ALIGNMENT) is the core of the proposed method, but the description is overly brief. To enhance readers' understanding, the authors could consider adding more technical details and method insights into the design choices. For instance, highlighting that different steps in model training focus on generating tokens for specific positions within the draft sequence would provide valuable context."**
>
> **A2:** Recall that the context inconsistency between training and decoding comes from the inaccuracies in features generated by the draft model compared to the features from the target LLM. The proposed method, i.e. harmonized context alignment, aims to address this inconsistency by adapting the inaccurate features in previous timesteps. It is achieved by first taking the inaccurate feature from the last draft model as query, and then considering the inaccuracy accumulation in key-value part of the transformer block.
> Formally, in the HASS training step $j$, given input token sequence $x_1, x_2, .., x_T$, we optimize the draft model $\mathcal{M}^{(s)}$ with the objective function
> $\min_{\mathcal{M}^{(s)}} \sum_{t=1}^{T-1} [\text{CrossEntropy}(P^{(l)}(x_{t+1}|x_1,\dots,x_t), P^{(s)}(x_{t+1}|x_1, \dots, x_t)) + \text{Aux-loss}],$
> where
> $$
> P^{(s)}(x_{t+1}|x_1, \dots, x_t) = \text{LM} \\ \text{Head}(f_{t+1}^{(s_j)})
> = \text{LM} \\ \text{Head}(
> \mathcal{M}^{(s)}(
> \overbrace{\underbrace{f_t^{(s_{j-1})}}_{\text{query}}}^{\text{From last draft}},
> \underbrace{
> \overbrace{f_1^{(l)} \oplus\cdots\oplus {f}\_{t-j+1}^{(l)}}\^{\text{From target LLM}}
> \oplus \overbrace{{f}\_{t-j+2}^{(s\_1)} \oplus \cdots \oplus {f}\_{t}^{(s\_{j-1})}}^{\text{From previous draft models}}}\_{\text{key}\\ \\& \\ \text{value}})),
> $$
> and $P^{(l)}$ is the auto-regressive probability distribution provided by the target LLM, and the $\text{Aux-loss}$ consists of the proposed Top-K loss and the feature regression loss (following EAGLE). When training tokens in the entire sequence in parallel, the above formulation adapts the inaccurate features in previous $j - 1$ steps for all positions except the first $j-1$ positions.
>
> To be more specifically, HASS training step 1 takes the target LLM's feature as input, and focuses on generating draft tokens in the draft model's first auto-regressive speculation during decoding; HASS training step 2 takes the target LLM's feature as well as the last step draft model's feature as input, and focuses on generating draft tokens in the draft model's second auto-regressive speculation during decoding; HASS training step $j$ takes the target LLM's feature as well as features from draft models in previous training steps as input, and focuses on generating draft tokens in the draft model's $j$-th auto-regressive speculation during decoding. Therefore, HASS adapts the inaccuracy accumulation from previous speculations during decoding.

---

> ### Author Response · Authors · 2024-11-20
> **Response to Reviewer ETQd (3)**
>
> **Q3: "What does the vertical axis of the line charts in Figures 5 and 6 represent? The ranges shown do not appear to correspond to the acceptance rate. If the authors are willing to address our concerns, we are open to raise the scores during the rebuttal stage."**
>
> **A3:** The vertical axis of the line charts in Figures 5 and 6 represents the acceptance rate $\alpha$, which has been transformed into percentage format.
> Since tree drafts sample multiple tokens per location with only one accepted, we follow EAGLE and utilize chain drafts without tree attention when measuring this metric, which is aligned with Speculative Sampling and Distillspec.

---

> ### Comment · Reviewer_ETQd · 2024-11-27
>
> Thank you for the detailed response, which effectively addresses my concerns. I have updated my score to 8 accordingly.

---

### Official Review · Reviewer_6oJb · 2024-11-05

**Soundness:** 4
**Presentation:** 2
**Contribution:** 3
**Rating:** 8
**Confidence:** 3

**Summary:**

The paper proposes two improvements to the speculative decoding method EAGLE. First, it regularizes training of the draft model to focus on top-K tokens. Second, it trains the draft model to use its own predictions when drafting more than one token. Both improvements are inspired by how speculative decoding is used during inference.

**Strengths:**

The paper carefully considers the two important aspects of speculative decoding efficiency: the ability of the draft model to predict top-K tokens and the number of tokens the draft model can successfully predict. Authors identify respective shortcomings of the recent method EAGLE and propose practical solutions.

- Empirical results are convincing and demonstrate meaningful improvements in terms of acceptance rate and speedups
- Authors present extensive ablation studies providing additional insights

**Weaknesses:**

A more rigorous mathematical presentation of the loss function in Section 3.2 (harmonized context alignment) would help to improve clarity. It is not easy to parse what is the input to the draft model at every step and what is it trying to predict. Writing out the objective function for training the draft model would help. I am also confused by Figure 3 - should Training Step 2 have $f^{(l)}\_{t-2}$ and $f^{(l)}\_{t-1}$ in the bottom row and only use superscript $(s_1)$ in the bottom right entry $f^{(s_1)}_{t}$ (analogously for steps 3 and 4)?

**Questions:**

- Could you please specify what data was used for training the draft model in the experiments? Does the proposed method (and EAGLE) suffer from generalization issues when speculative decoding is tested on data different from what was used to train the draft model?
- Can training on Model Generated data mitigate the context misalignment? In other words, will HASS outperform EAGLE-2 trained on Model-Generated data? It'd be interesting to see a comparison to EAGLE-2 in Table 8.

My rating for this paper is (7), which is not available as an option. If my questions and clarity concerns are addressed, I will be happy to consider raising the rating to (8).

---

> ### Author Response · Authors · 2024-11-20
> **Response to Reviewer 6oJb (1)**
>
> **Thanks for your valuable comments.**
>
> **Q1: "A more rigorous mathematical presentation of the loss function in Section 3.2 (harmonized context alignment) would help to improve clarity. It is not easy to parse what is the input to the draft model at every step and what is it trying to predict. Writing out the objective function for training the draft model would help."**
>
> **A1:** Recall that the context inconsistency between training and decoding comes from the inaccuracies in features generated by the draft model compared to the features from the target LLM. The proposed method, i.e. harmonized context alignment, aims to address this inconsistency by adapting the inaccurate features in previous timesteps. It is achieved by first taking the inaccurate feature from the last draft model as query, and then considering the inaccuracy accumulation in key-value part of the transformer block.
> Formally, in the HASS training step $j$, given input token sequence $x_1, x_2, .., x_T$, we optimize the draft model $\mathcal{M}^{(s)}$ with the objective function
>
> $\min_{\mathcal{M}^{(s)}} \sum_{t=1}^{T-1} [\text{CrossEntropy}(P^{(l)}(x_{t+1}|x_1,\dots,x_t), P^{(s)}(x_{t+1}|x_1, \dots, x_t)) + \text{Aux-loss}],$
>
> where
> $$
> P^{(s)}(x_{t+1}|x_1, \dots, x_t) = \text{LM} \\ \text{Head}(f_{t+1}^{(s_j)})
> = \text{LM} \\ \text{Head}(
> \mathcal{M}^{(s)}(
> \overbrace{\underbrace{f_t^{(s_{j-1})}}_{\text{query}}}^{\text{From last draft}},
> \underbrace{
> \overbrace{f_1^{(l)} \oplus\cdots\oplus {f}\_{t-j+1}^{(l)}}\^{\text{From target LLM}}
> \oplus \overbrace{{f}\_{t-j+2}^{(s\_1)} \oplus \cdots \oplus {f}\_{t}^{(s\_{j-1})}}^{\text{From previous draft models}}}\_{\text{key}\\ \\& \\ \text{value}})),
> $$
>
>
> and $P^{(l)}$ is the auto-regressive probability distribution provided by the target LLM, and the $\text{Aux-loss}$ consists of the proposed Top-K loss and the feature regression loss (following EAGLE). When training tokens in the entire sequence in parallel, the above formulation adapts the inaccurate features in previous $j - 1$ steps for all positions except the first $j-1$ positions.

---

> ### Author Response · Authors · 2024-11-20
> **Response to Reviewer 6oJb (2)**
>
> **Q2: "I am also confused by Figure 3 - should Training Step 2 have $f_{t-2}^{(l)}$ and $f_{t-1}^{(l)}$ in the bottom row and only use superscript $(s_1)$ in the bottom right entry $f_{t}^{(s_1)}$ (analogously for steps 3 and 4)?"**
>
> **A2:** In HASS training step 2, if we only use superscript $(s_1)$ in the bottom right entry $f_t^{(s_1)}$ and use superscript $(l)$ for the others, merely the last position of the input sequence can adapt to the inaccurate feature generated by last draft, while the rest positions degenerate to HASS training step 1. Oppositely, if we use superscript $(s_1)$ for the last three entries in the bottom row, the entire sequence can adapt to the inaccurate features generated by last draft except for the first position.

---

> ### Author Response · Authors · 2024-11-20
> **Response to Reviewer 6oJb (3)**
>
> **Q3: "Could you please specify what data was used for training the draft model in the experiments? Does the proposed method (and EAGLE) suffer from generalization issues when speculative decoding is tested on data different from what was used to train the draft model?"**
>
> **A3:** In all the experiments, we use the fixed ShareGPT dataset with 68,000 dialogues for training draft models by following [B, C, D], unless stated otherwise.
>
> We evaluate HASS and EAGLE-2 trained from ShareGPT on dialogue (MT-bench), code generation (HumanEval), and mathematical reasoning (GSM8K) tasks in the main text. We also evaluate HASS and EAGLE-2 trained from ShareGPT on translation tasks following [A] in Appendix A.7 Evaluation on Translation Tasks.
>
> We show the speedup ratios and acceptance lengths $\tau$ of HASS and EAGLE-2 averaged across MT-bench, HumanEval, and GSM8K (MT+HE+GSM) and averaged across five translation tasks (Translation) in the following table.
>
> |                     |         | Temp=0    |          |             |          | Temp=1    |          |             |          |
> | ------------------- | ------- | --------- | -------- | ----------- | -------- | --------- | -------- | ----------- | -------- |
> |                     |         | MT+HE+GSM |          | Translation |          | MT+HE+GSM |          | Translation |          |
> | Model               | Method  | Speedup   | $\tau$   | Speedup     | $\tau$   | Speedup   | $\tau$   | Speedup     | $\tau$   |
> | LLaMA2-Chat 7B      | EAGLE-2 | 2.81x     | 4.61     | 2.43x       | 3.80     | 2.60x     | 4.40     | 2.20x       | 3.66     |
> |                     | HASS    | **3.24x** | **5.15** | **2.92x**   | **4.27** | **2.90x** | **4.92** | **2.73x**   | **4.19** |
> | LLaMA2-Chat 13B     | EAGLE-2 | 3.30x     | 5.16     | 2.77x       | 4.06     | 3.21x     | 5.01     | 2.60x       | 3.91     |
> |                     | HASS    | **3.65x** | **5.58** | **3.28x**   | **4.45** | **3.48x** | **5.42** | **3.06x**   | **4.37** |
> | LLaMA3-Instruct 8B  | EAGLE-2 | 2.83x     | 4.52     | 2.15x       | 3.45     | 2.47x     | 4.31     | 2.02x       | 3.27     |
> |                     | HASS    | **3.09x** | **5.08** | **2.52x**   | **3.72** | **2.81x** | **4.80** | **2.44x**   | **3.65** |
> | LLaMA3-Instruct 70B | EAGLE-2 | 3.37x     | 4.50     | 2.78x       | 3.71     | 3.28x     | 4.43     | 2.69x       | 3.61     |
> |                     | HASS    | **4.05x** | **5.21** | **3.25x**   | **4.12** | **3.85x** | **5.16** | **3.21x**   | **4.07** |
>
> Both HASS and EAGLE-2 demonstrate generalization ability across tasks different from the training task, while HASS consistently outperforms EAGLE-2 with a promising improvement.
>
> [A] Yi et al., 2024. Towards Fast Multilingual LLM Inference: Speculative Decoding and Specialized Drafters, EMNLP 2024.
>
> [B] Li et al., 2024. EAGLE: Speculative Sampling Requires Rethinking Feature Uncertainty, ICML 2024.
>
> [C] Cai et al., 2024. MEDUSA: Simple LLM Inference Acceleration Framework with Multiple Decoding Heads, ICML 2024.
>
> [D] Du et al., 2024. GLIDE with a CAPE: A Low-Hassle Method to Accelerate Speculative Decoding, ICML 2024.

---

> ### Author Response · Authors · 2024-11-20
> **Response to Reviewer 6oJb (4)**
>
> **Q4: "Can training on Model Generated data mitigate the context misalignment? In other words, will HASS outperform EAGLE-2 trained on Model-Generated data? It'd be interesting to see a comparison to EAGLE-2 in Table 8."**
>
> **A4:** We train EAGLE-2 on the model-generated data and report its comparison with HASS in Table 8 Appendix A.4 Self-Distillation.
> We show the speedup ratios and acceptance lengths $\tau$ of HASS and EAGLE-2 trained with fixed and model-generated data averaged across MT-bench, HumanEval, and GSM8K in the following table.
>
> |                 |         |      | Temp=0                             |                                   | Temp=1                             |                                   |
> | --------------- | ------- | ---- | ---------------------------------- | --------------------------------- | ---------------------------------- | --------------------------------- |
> | Model           | Method  | Data | Speedup                            | $\tau$                            | Speedup                            | $\tau$                            |
> | LLaMA2-Chat 7B  | EAGLE-2 | F    | 2.81x                              | 4.61                              | 2.60x                              | 4.40                              |
> |                 |         | MG   | **3.06x** ($\color{green}{+0.25}$) | **4.94** ($\color{green}{+0.33}$) | **2.71x** ($\color{green}{+0.11}$) | **4.64** ($\color{green}{+0.24}$) |
> |                 | HASS    | F    | 3.24x                              | 5.15                              | 2.90x                              | 4.92                              |
> |                 |         | MG   | **3.46x** ($\color{green}{+0.22}$) | **5.51** ($\color{green}{+0.36}$) | **3.09x** ($\color{green}{+0.19}$) | **5.19** ($\color{green}{+0.27}$) |
> | LLaMA2-Chat 13B | EAGLE-2 | F    | **3.30x**                          | **5.16**                          | **3.21x**                          | **5.01**                          |
> |                 |         | MG   | 3.23x ($\color{red}{-0.07}$)       | 5.14 ($\color{red}{-0.02}$)       | 3.12x ($\color{red}{-0.09}$)       | 4.94 ($\color{red}{-0.07}$)       |
> |                 | HASS    | F    | 3.65x                              | 5.58                              | 3.48x                              | **5.42**                          |
> |                 |         | MG   | **3.80x** ($\color{green}{+0.15}$) | **5.63** ($\color{green}{+0.05}$) | **3.56x** ($\color{green}{+0.08}$) | 5.41 ($\color{red}{-0.01}$)       |
>
> As shown from the table, HASS outperforms EAGLE-2 on either fixed training data or model-generated training data. It is noted that HASS trained on the fixed dataset even achieves better performance than EAGLE-2 trained on the model-generated data consistently. With model-generated data for training, HASS consistently achieves more improvement or less degeneration in terms of the acceptance length compared with EAGLE-2. Therefore, context misalignment is not mitigated by training on model-generated data.

---

> > ### Comment · Reviewer_6oJb · 2024-11-27
> >
> > Thank you for the responses and additional experimental results. Please update the paper to include additional technical details regarding the method. I've revised my score.

---

### Official Review · Reviewer_RwqU · 2024-11-08

**Soundness:** 3
**Presentation:** 3
**Contribution:** 3
**Rating:** 6
**Confidence:** 5

**Summary:**

Hass method includes leveraging a hybrid autoregressive and non-autoregressive (HASS) approach to optimize decoding time. The authors present experimental results demonstrating improved performance compared to baseline SD methods.

**Strengths:**

The experiments are well-detailed, with clear metrics for comparison against baselines (e.g., EAGLE and multiple architectures).

The method makes sense in the perspective of typical acceptance process in SD.

**Weaknesses:**

1. While the paper introduces a novel approach, it does not sufficiently explore the construction and utilization of the self-distillation dataset. Isn't the quality and configuration of this dataset more crucial than the framework itself? A deeper discussion on how the dataset is designed and its influence on model performance would strengthen the claims [Referring to A].

2. The paper lacks experiments analyzing how the framework performs across different token counts and task types [Referring to A]. Adding these analyses could provide a more comprehensive understanding of the proposed method's scalability and robustness. For instance, how does performance vary when the token count significantly increases or decreases? Are there certain task types where the proposed method excels or struggles compared to baselines?

[A] Yi et al., 2024. Towards Fast Multilingual LLM Inference: Speculative Decoding and Specialized Drafters, EMNLP 2024-main.

**Questions:**

Seek weakness.

Will update the score after looking at the results of Weakness.

---

> ### Author Response · Authors · 2024-11-20
> **Response to Reviewer RwqU (1)**
>
> **Thanks for your valuable comments.**
>
> **Q1: "While the paper introduces a novel approach, it does not sufficiently explore the construction and utilization of the self-distillation dataset. Isn't the quality and configuration of this dataset more crucial than the framework itself? A deeper discussion on how the dataset is designed and its influence on model performance would strengthen the claims [Referring to A]."**
>
> **A1:** It is worth mentioning that we only use the fixed ShareGPT dataset (https://huggingface.co/datasets/Aeala/ShareGPT_Vicuna_unfiltered) to train draft models for all the experiments unless stated otherwise, which is aligned with the settings of [B, C, D].
> We also contruct self-distilled ShareGPT dataset by feeding the prompts from ShareGPT into the target LLM recursively with temperature set to 0 and collecting the responses as multi-turn conversations following [B], and summarize the results of HASS and EAGLE-2 trained on the self-distillation dataset in Appendix A.4 Self-Distillation.
>
> We show the speedup ratios and acceptance lengths $\tau$ of HASS and EAGLE-2 averaged across MT-bench, HumanEval, and GSM8K with fixed and self-distilled ShareGPT as training dataset in the following table, where F and MG stand for Fixed and Model-Generated, respectively.
>
> |                 |         |      | Temp=0                             |                                   | Temp=1                             |                                   |
> | --------------- | ------- | ---- | ---------------------------------- | --------------------------------- | ---------------------------------- | --------------------------------- |
> | Model           | Method  | Data | Speedup                            | $\tau$                            | Speedup                            | $\tau$                            |
> | LLaMA2-Chat 7B  | EAGLE-2 | F    | 2.81x                              | 4.61                              | 2.60x                              | 4.40                              |
> |                 |         | MG   | **3.06x** ($\color{green}{+0.25}$) | **4.94** ($\color{green}{+0.33}$) | **2.71x** ($\color{green}{+0.11}$) | **4.64** ($\color{green}{+0.24}$) |
> |                 | HASS    | F    | 3.24x                              | 5.15                              | 2.90x                              | 4.92                              |
> |                 |         | MG   | **3.46x** ($\color{green}{+0.22}$) | **5.51** ($\color{green}{+0.36}$) | **3.09x** ($\color{green}{+0.19}$) | **5.19** ($\color{green}{+0.27}$) |
> | LLaMA2-Chat 13B | EAGLE-2 | F    | **3.30x**                          | **5.16**                          | **3.21x**                          | **5.01**                          |
> |                 |         | MG   | 3.23x ($\color{red}{-0.07}$)       | 5.14 ($\color{red}{-0.02}$)       | 3.12x ($\color{red}{-0.09}$)       | 4.94 ($\color{red}{-0.07}$)       |
> |                 | HASS    | F    | 3.65x                              | 5.58                              | 3.48x                              | **5.42**                          |
> |                 |         | MG   | **3.80x** ($\color{green}{+0.15}$) | **5.63** ($\color{green}{+0.05}$) | **3.56x** ($\color{green}{+0.08}$) | 5.41 ($\color{red}{-0.01}$)       |
>
> Self-distillation consistently brings improvements for HASS and EAGLE-2 on LLaMA2-Chat 7B, whille it only achieves marginally better or comparable results on LLaMA2-Chat 13B, which is consistent with the observation from [B] ("data from the target LLM marginally improves performance" in [B] section 4.3.3). This may be due to the code generation dataset HumanEval and the mathematical reasoning dataset GSM8K being less similar to the training dataset ShareGPT compared with MT-bench.
> There could be more ways to construct the self-distillation dataset such as varying the temperature during generation, but they are out of the scope of this paper. Our main focus is addressing the disharmonies between the draft model's training and decoding on their objectives and contexts, which yields an auto-regressive draft model having better alignment with the target LLM. With affordable training overhead, HASS requires no extra inference overhead compared to EAGLE-2 and consistently outperforms exisiting baselines.
>
> [A] Yi et al., 2024. Towards Fast Multilingual LLM Inference: Speculative Decoding and Specialized Drafters, EMNLP 2024.
>
> [B] Li et al., 2024. EAGLE: Speculative Sampling Requires Rethinking Feature Uncertainty, ICML 2024.
>
> [C] Cai et al., 2024. MEDUSA: Simple LLM Inference Acceleration Framework with Multiple Decoding Heads, ICML 2024.
>
> [D] Du et al., 2024. GLIDE with a CAPE: A Low-Hassle Method to Accelerate Speculative Decoding, ICML 2024.

---

> ### Author Response · Authors · 2024-11-20
> **Response to Reviewer RwqU (2)**
>
> **Q2: "The paper lacks experiments analyzing how the framework performs across different token counts ... [Referring to A]. Adding these analyses could provide a more comprehensive understanding of the proposed method's scalability and robustness. For instance, how does performance vary when the token count significantly increases or decreases?"**
>
> **A2:** To investigate the influences of training token numbers as inspired by [A], we randomly sample different proportions of the training dataset, i.e., the ShareGPT dataset with 68,000 dialogues. In specific, we train draft models of HASS and EAGLE-2 with 1/8, 1/4, 1/2 and the entire ShareGPT dataset and summarize the results in Appendix A.6 Number of Training Tokens.
>
> We show the speedup ratios and acceptance lengths $\tau$ of HASS and EAGLE-2 averaged across MT-bench, HumanEval, and GSM8K with different proportions of ShareGPT as training dataset in the following table.
>
> |                    |         |            | Temp=0    |          | Temp=1    |          |
> | ------------------ | ------- | ---------- | --------- | -------- | --------- | -------- |
> | Model              | Method  | Proportion | Speedup   | $\tau$   | Speedup   | $\tau$   |
> | LLaMA2-Chat 7B     | EAGLE-2 | 1/8        | 1.95x     | 3.23     | 1.79x     | 3.12     |
> |                    |         | 1/4        | 2.26x     | 3.79     | 2.15x     | 3.63     |
> |                    |         | 1/2        | 2.59x     | 4.28     | 2.43x     | 4.08     |
> |                    |         | 1/1        | **2.81x** | **4.61** | **2.60x** | **4.40** |
> |                    | HASS    | 1/8        | 2.52x     | 4.11     | 2.37x     | 3.97     |
> |                    |         | 1/4        | 2.80x     | 4.59     | 2.65x     | 4.42     |
> |                    |         | 1/2        | 3.13x     | 4.93     | 2.85x     | 4.71     |
> |                    |         | 1/1        | **3.24x** | **5.15** | **2.90x** | **4.92** |
> | LLaMA2-Chat 13B    | EAGLE-2 | 1/8        | 2.18x     | 3.54     | 2.11x     | 3.44     |
> |                    |         | 1/4        | 2.67x     | 4.20     | 2.60x     | 4.06     |
> |                    |         | 1/2        | 3.09x     | 4.77     | 3.02x     | 4.62     |
> |                    |         | 1/1        | **3.30x** | **5.16** | **3.21x** | **5.01** |
> |                    | HASS    | 1/8        | 2.99x     | 4.41     | 2.82x     | 4.32     |
> |                    |         | 1/4        | 3.22x     | 5.02     | 3.13x     | 4.88     |
> |                    |         | 1/2        | 3.59x     | 5.36     | 3.39x     | 5.21     |
> |                    |         | 1/1        | **3.65x** | **5.58** | **3.48x** | **5.42** |
> | LLaMA3-Instruct 8B | EAGLE-2 | 1/8        | 1.72x     | 2.91     | 1.59x     | 2.82     |
> |                    |         | 1/4        | 2.09x     | 3.37     | 1.85x     | 3.26     |
> |                    |         | 1/2        | 2.43x     | 3.92     | 2.18x     | 3.75     |
> |                    |         | 1/1        | **2.83x** | **4.52** | **2.47x** | **4.31** |
> |                    | HASS    | 1/8        | 2.43x     | 3.87     | 2.17x     | 3.75     |
> |                    |         | 1/4        | 2.73x     | 4.38     | 2.39x     | 4.19     |
> |                    |         | 1/2        | 3.03x     | 4.81     | 2.74x     | 4.58     |
> |                    |         | 1/1        | **3.09x** | **5.08** | **2.81x** | **4.80** |
> | LLaMA3-Instruct 70B | EAGLE-2 | 1/8        | 2.37x     | 3.08     | 2.29x     | 3.04     |
> |                    |         | 1/4        | 2.81x     | 3.60     | 2.61x     | 3.56     |
> |                    |         | 1/2        | 3.19x     | 4.09     | 3.02x     | 4.03     |
> |                    |         | 1/1        | **3.37x** | **4.50** | **3.28x** | **4.43** |
> |                    | HASS    | 1/8        | 3.21x     | 4.08     | 3.08x     | 4.05     |
> |                    |         | 1/4        | 3.61x     | 4.61     | 3.47x     | 4.57     |
> |                    |         | 1/2        | 3.89x     | 4.96     | 3.72x     | 4.91     |
> |                    |         | 1/1        | **4.05x** | **5.21** | **3.85x** | **5.16** |
>
>
> As shown from the table, HASS consistently outperforms EAGLE-2 under different proportions of training dataset with temperature $\in \\{ 0,1 \\}$. HASS with merely 1/4 training dataset achieves better or comparable performance compared to EAGLE-2 with the entire training dataset, which demonstrates HASS’s superior data exploitation and scalability obtained through further aligning on objectives and contexts between training and decoding. The speedup ratio and acceptance length of HASS and EAGLE-2 are approximately logarithmically proportional to the scale of training data, which is consistent with the finding in [A], and so we are happy to cite [A] in our paper.

---

> ### Author Response · Authors · 2024-11-20
> **Response to Reviewer RwqU (3)**
>
> **Q3: "The paper lacks experiments analyzing how the framework performs across different ... task types [Referring to A]. Adding these analyses could provide a more comprehensive understanding of the proposed method's scalability and robustness. Are there certain task types where the proposed method excels or struggles compared to baselines?"**
>
> **A3:** In the main text, HASS is trained on the fixed ShareGPT dataset with 68,000 dialogues and evaluated on three different task types, i.e., dialogue (MT-bench), code generation (HumanEval), and mathematical reasoning (GSM8K), where the proposed HASS consistently outperforms other baselines.
>
> In Appendix A.7 Evaluation on Translation Tasks, we further evaluate HASS and EAGLE-2 trained from the fixed ShareGPT dataset on five translation tasks (https://github.com/Kthyeon/Multilingual-SpecBench/tree/main/dataset/eval) by following [A].
>
> We show the speedup ratios and acceptance lengths $\tau$ of HASS and EAGLE-2 averaged across five translation tasks, i.e., De$\rightarrow$En, Fr$\rightarrow$En, Ja$\rightarrow$En, Ru$\rightarrow$En, Zh$\rightarrow$En, in the following table. It is noted that De, Fr, Ja, Ru, Zh, and En stand for German, French, Japanese, Russian, Chinese, and English, respectively. Please refer to Table 11 in Appendix A.7 for results on each translation task.
>
> |                     |         | Temp=0    |          | Temp=1    |          |
> | ------------------- | ------- | --------- | -------- | --------- | -------- |
> | Model               | Method  | Speedup   | $\tau$   | Speedup   | $\tau$   |
> | LLaMA2-Chat 7B      | EAGLE-2 | 2.43x     | 3.80     | 2.20x     | 3.66     |
> |                     | HASS    | **2.92x** | **4.27** | **2.73x** | **4.19** |
> | LLaMA2-Chat 13B     | EAGLE-2 | 2.77x     | 4.06     | 2.60x     | 3.91     |
> |                     | HASS    | **3.28x** | **4.45** | **3.06x** | **4.37** |
> | LLaMA3-Instruct 8B  | EAGLE-2 | 2.15x     | 3.45     | 2.02x     | 3.27     |
> |                     | HASS    | **2.52x** | **3.72** | **2.44x** | **3.65** |
> | LLaMA3-Instruct 70B | EAGLE-2 | 2.78x     | 3.71     | 2.69x     | 3.61     |
> |                     | HASS    | **3.25x** | **4.12** | **3.21x** | **4.07** |
>
> As shown from the table, HASS consistently outperforms EAGLE-2 under all settings. HASS achieves 2.44x-3.28x wall-clock time speedup ratio averaging across five translation tasks, surpassing EAGLE-2 by 17%-24%. In terms of acceptance length, HASS achieves 8%-14% improvement over EAGLE-2. In consistent with results on MT-bench, HumanEval, and GSM8K, HASS shows promising improvements over EAGLE-2 on translation tasks, reflecting its robustness across different task types.

---

> > ### Comment · Reviewer_RwqU · 2024-11-25
> >
> > The authors have addressed all concerns thoroughly with detailed experiments. While some limitations remain, their responses align well with the paper's focus and scope. Raising the score to 6 is reasonable.

---

### Author Response · Authors · 2024-11-25
**Appreciating Your Feedback**

We would appreciate your feedback if any concerns remain as the discussion phase comes to a close. Thank all the reviewers for your time and we are grateful for the discussion opportunity.

Thanks!

---

### Meta-Review · Area_Chair_Ubjt · 2024-12-16

**Metareview:**

The paper proposes an approach to enhance the speedup offered by speculative decoding. The idea is to reduce the discrepancy between drafter training and inference, particularly for the recent EAGLE series of methods. This involves employing a ranking-based distillation loss, and a training loss that incorporates the drafter's autoregressively generated features as additional context (to complement the verifier's features, as in regular EAGLE). These are shown to yield consistent gains over the EAGLE series of methods.

Reviewers were unanimously supportive of the paper. Speculative decoding is widely used to improve LLMs' inference speed, and further enhancing this technique is of clear interest. The core idea of reducing train-inference discrepancy is sensible, and the paper's empirical results are impressive.

Reviewers did point out one drawback of method being the increased training time, although the authors empirically demonstrated that it is not exorbitant (and has no impact on the decoding process). There were also suggestions on improving the clarity of presentation, which in the AC's reading could be further enhanced (see below). Overall, however, the paper's contribution is expected to be useful to the field.

_Suggestions for presentation_:
- Figure 1, font sizes can be increased
- Figure 3, several things are not clear. What do the dashed arrows emanating from $\mathcal{M}_s$ represent? Presumably the rectangular array on the right is the attention matrix; but why is there a single label for $q^{(s_i)}$? What is the significance of the colours changing across the timesteps?

**Additional Comments On Reviewer Discussion:**

Initial reviews were unanimously positive. Reviewers suggested that the description of the method could be clearer, which was mostly addressed in the revision following the author response. There were also questions regarding the distillation process, which were clarified.

---

### Decision · Program_Chairs · 2025-01-22

Accept (Poster)